# Does Writing with Language Models Reduce Content Diversity?

**Vishakh Padmakumar**
New York University
vishakh@nyu.edu

**He He**
New York University
hehe@cs.nyu.edu

## Abstract

Large language models (LLMs) have led to a surge in collaborative writing with model assistance. As different users incorporate suggestions from the same model, there is a risk of decreased diversity in the produced content, potentially limiting diverse perspectives in public discourse. In this work, we measure the impact of co-writing on diversity via a controlled experiment, where users write argumentative essays in three setups—using a base LLM (GPT3), a feedback-tuned LLM (InstructGPT), and writing without model help. We develop a set of diversity metrics and find that writing with InstructGPT (but not the GPT3) results in a statistically significant reduction in diversity. Specifically, it increases the similarity between the writings of different authors and reduces the overall lexical and content diversity. We additionally find that this effect is mainly attributable to InstructGPT contributing less diverse text to co-written essays. In contrast, the user-contributed text remains unaffected by model collaboration. This suggests that the recent improvement in generation quality from adapting models to human feedback might come at the cost of more homogeneous and less diverse content.

## 1 Introduction

Large language models (LLMs) are rapidly changing how people create content (Lee et al., 2022a; Mirowski et al., 2023). While LLM-based writing assistants have the potential to improve writing quality and increase author productivity, they also introduce an algorithmic monoculture (Kleinberg and Raghavan, 2021). As millions of users rely on the same underlying model to produce text, there is a potential risk of shifting the content towards the mode—good but not colorful writing. In this work, we aim to assess whether writing with LLMs unintentionally reduces content diversity.

Existing evidence has already hinted that LLMs may influence users' opinions. Jakesch et al. (2023) and Bhat et al. (2023) find that users' opinions expressed in their writing can be biased by controlling LLMs to promote a specific argument (e.g., social media is good/bad for society). Additionally, it has been shown that current LLMs do not equally represent views from various demographic groups (Santurkar et al., 2023; Durmus et al., 2023). These results suggest that writing with LLMs may limit the perspectives expressed in the writing. We hypothesize that incorporating model suggestions dilutes the writer's unique voice, leading to different writers producing similar content, and this homogenization in turn reduces the overall diversity of content produced by many writers.

To test our hypotheses, we design a controlled experiment (Figure 1) where users are asked to write an argumentative essay given a topic from the New York Times student opinion series following Lee et al. (2022a), e.g., "What are the most important things students learn at school?" We assign participants to three groups: a *control group* where participants write essays without model help, an *LLM treatment group* where participants write essays with a base language model (GPT3), and a *feedback-tuned LLM treatment group* where participants write essays with a language model finetuned with human feedback (Ouyang et al., 2022) (InstructGPT). We hired 38 writers from Upwork. For each group, we collected 100 essays on 10 topics. We then develop a set of metrics and measure the effect of LLMs on content diversity at both the individual level and the collective level:

- **Homogenization:** *Do users write more similarly to each other when writing with LLMs?*

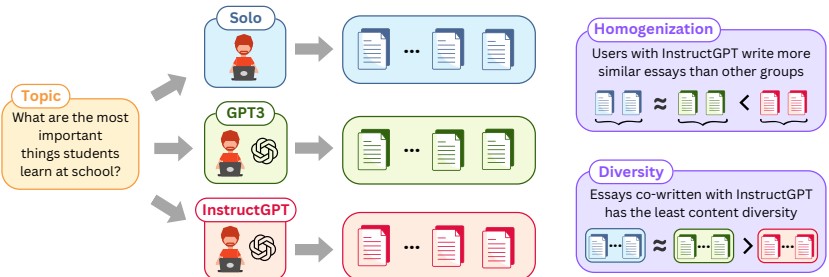

Figure 1: We measure the content diversity of essays written by three groups of users: a control group writing without model help (Solo), a treatment group writing with a base language model (GPT3), and second treatment group writing with a feedback-tuned language model (InstructGPT). Essays co-written with InstructGPT are more similar to each other and exhibit the least lexical and content diversity, whereas those co-written with GPT3 are not significantly different to the Solo essays.

We find that essays written by the group using InstructGPT exhibit a higher degree of homogenization compared to the control group and the GPT3 group (Section 4). In particular, by matching model-contributed text to the summarized key points of each essay, we find that the key points contributed by InstructGPT result in increased homogenization (Section 6).

- **Diversity:** *Does writing with LLMs reduce the diversity of content produced by a group of users?*

  We find that the set of essays written with InstructGPT show both lower lexical diversity and reduced diversity in the key points expressed as compared to the other two groups (Section 5). This is seen in the increased repetition of phrases (higher-order $n$-grams) introduced by the model (Section 5.2) and a reduction in the number of unique key points in the essays.

Interestingly, while writing with the human feedback-tuned LLM (InstructGPT) incurs increased homogenization and reduced diversity, we did not observe a statistically significant effect from writing with the base LLM (GPT3) despite a similar model contribution (Section 3). Further analysis shows that InstructGPT tends to provide less diverse suggestions and introduce less diverse text compared to GPT3. This finding echoes prior results showing reduced diversity after reinforcement learning from human feedback (Bai et al., 2022; Song et al., 2023).

As LLMs become more tightly integrated into text editing applications, our findings highlight a new axis of evaluation in interactive settings. Reduced content diversity is not only detrimental to personal expression and creativity, but also risks creating a feedback loop as future models trained on the homogenized content further propagate this pattern (Taori and Hashimoto, 2023). To facilitate future research on understanding the impact of LLMs in co-writing, we release essays from all three groups along with all model suggestions recorded by the CoAuthor platform (Lee et al., 2022a).[1]

## 2 DATA COLLECTION

Our approach to studying the impact of LLMs on content diversity is to conduct controlled experiments. Specifically, we control the set of writers and the topics of the essays. We then compare essays written with and without model help.

### 2.1 TASK SETUP

We consider the task of argumentative essay writing using topics from the New York Times Student Opinion series following Lee et al. (2022a). Specifically, users are given a topic such as "What are the most important things students learn at school?" and are asked to write an essay expressing their opinions in around 300 words. We choose this task because the topics are sufficiently open-ended to allow for diverse responses while remaining accessible to users with all backgrounds. The instructions and the complete list of 10 prompts used in the experiment are provided in Appendix A.3.

---

[1]Code and data are available at `https://github.com/vishakhpk/hai-diversity`.

We adopt the CoAuthor platform developed by Lee et al. (2022a). When the user hits the TAB key, we obtain 5 text continuations from a backend model and present them in a dropdown menu. Figure 5 in Appendix A shows a screenshot of the interface as the user requests model suggestions. The user then has the option to either accept and edit one of the suggestions or reject all suggestions and continue with their writing process. We ask the users to request suggestions a minimum of 5 times per essay but do not require them to accept any of these. The entire writing process, including all keystrokes, suggestions, and user actions, is recorded by the platform. This allows us to differentiate the parts of the essay introduced by the users and models at the character level for subsequent analysis.

## 2.2 EXPERIMENT SETUP

We recruit participants from Upwork who are native English speakers and have prior experience in writing or copyediting on the platform. We provide further details on participant recruitment and remuneration in Appendix A.1. We then collect essays from three settings: a control group, where participants write essays without any model help; an LLM treatment group, where participants write with the help of a base language model; and a feedback-tuned LLM treatment group, where participants write with a language model finetuned on human feedback. We distinguish the two types of language models because prior work has shown that finetuning language models on human feedback tends to decrease the entropy of the output distribution Bai et al. (2022) and thus may decrease text diversity in general. We henceforth refer to the three settings as **Solo**, **GPT3**, and **InstructGPT** respectively.

We obtain model continuations from the OpenAI API using `davinci` as the base language model and `text-davinci-003` as the feedback-tuned language model. To avoid reduced diversity due to decoding, we use a decoding strategy biased towards higher entropy. Specifically, we sample continuations from both models with a temperature of $0.9$ and a frequency penalty of $0.5$ (detailed parameters listed in Appendix A.2), following the "high randomness" setting from Lee et al. (2022a).

Each participant is asked to complete a session consisting of three essay writing tasks—one in each of the three settings and each on a different topic. In each session, the order of the three settings and the assignment of topics are randomized. Each essay takes 15 minutes on average. In total, we obtained 10 essays on each of the 10 topics for each setting, resulting in 300 essays in total.

## 3 HOW MUCH DO USERS ENGAGE WITH THE MODEL?

Before diving into the analysis of content diversity, we first examine whether the user and the language models form an effective collaboration by examining the usage statistics and model-contributed text.

| Model | # Queries | Acceptance Rate (%) | Model-Written Percentage | Word Count |
|---|---|---|---|---|
| InstructGPT | 9.15 | 70.49 | 32.45 | 368.39 |
| GPT3 | 9.62 | 71.32 | 35.57 | 380.87 |

Table 1: Averaged per-essay usage statistics of GPT3 and InstructGPT. Both models are helpful to users, evidenced by the high acceptance rates of suggestions and the sizeable model written fraction of essays. Furthermore, we find no statistically significant difference in usage of the two models.

**Usage statistics.** We find that users actively query the model and use model suggestions during writing. As shown in Table 1, users query the model around 9 times on average for each essay and accept about $70\%$ of them (note that they are asked to query at least 5 times). Since users may further edit the suggestions after accepting them, we want to know whether the accepted suggestions are retained in the final essay.[2] The interface used for data collection records keystroke-level information about whether each character was introduced by the model or the user. Thus, we report the average percentage of characters introduced by the model in each essay. We observe that the model contributes roughly $35\%$ of the characters in each essay, suggesting that they are reasonably helpful to the writing task. Finally, we are curious whether the users find InstructGPT to be more helpful than GPT3 given its additional feedback tuning. However, the two language models appear to be equally helpful to

---

[2]Participants are not obligated to accept any suggestions from the model. We ask that they query the model for suggestions at least 5 times when writing an essay, accepting these only when appropriate (Appendix A.3).

While I believe the concerns regarding children's screen time are valid, **I believe it is somewhat biased to not take this problem, which is a genuine issue right now, as an everyone problem**, `[skipped]` I know that I, along with many other teenagers, would like nothing more than to go back to school, play sports outside, meet new people, and such. `[skipped]` **They are also places for new ideas, watching college lectures, and political discourse** `[skipped]`

- The problem of screen time should be considered an everyone problem, not just a student one
- Social media can be used for educational purposes
- Limiting screen time may not be effective in the long run
- Parents should trust their teenager more and not worry too much about their screen time

Table 2: Excerpt from an essay written by a user with InstructGPT and the generated key points. Bold text was introduced by the model. We align each key point to a sentence in the essay with which it has the highest Rouge-L overlap, as illustrated by the highlight. Since the majority of the text in the aligned sentence is written by the model, we attribute the first key point to the model.

users. We do not observe a significant difference between their usage statistics: an independent samples $t$-test on the number of queries and acceptance rates shows no significance at the $5\%$ level.

**Model contribution to key ideas.** The usage statistics tell us that LLMs contribute to a fair amount of the text written. However, are they contributing key arguments or merely supporting the elaboration of a point? To quantify model contribution to the main ideas, we need a way to represent the important content in an essay. Motivated by recent encouraging results of using LLMs for zero-shot summarization (Goyal et al., 2022), we summarize each essay into a list of key points by prompting `gpt-3.5-turbo`.[3] An example is shown in Table 2.[4]

Given a list of key points for each essay, we then estimate the fraction of key points written by the model and the user. Specifically, we align each key point to a sentence in the essay with which it has the highest overlap, as measured by Rouge-L (Lin, 2004). If more than half of the sentence's characters are model-generated based on the recorded keystroke data, the key point is attributed to the model; otherwise, it is attributed to the user. We show the fraction of key points written by the model for the two groups of co-written essays in Figure 7 in Appendix C. We observe that the median fraction for both GPT3 and InstructGPT is around $0.4$, indicating that models make a significant contribution to the key content of the essay. However, there is also a large variation among users, with some essays having none or all of the key points attributed to the model, suggesting diverse levels of reliance on the model.

Given the significant model contribution in co-written essays, we hypothesize that this intervention would result in increased content similarity among users. We test this hypothesis in Section 4.

## 4 DOES WRITING WITH LLMS RESULT IN MORE SIMILAR ESSAYS?

In the previous section, we have seen that models contribute a substantial portion of the essays' content. Does this lead to more similar essays, or do the essays retain the individual styles of the users? In this section, we measure the homogenization effect of co-writing with LLMs.

### 4.1 METRIC

We first define the homogenization of a single essay as its average pairwise similarity to all other essays written on the same topic. Let $D_t$ denote the set of essays on topic $t$. The homogenization score of an essay $d$ (written in response to topic $t$) is $\text{hom}(d \mid t) = \frac{1}{|D_t|-1} \sum_{d' \in D_t \setminus d} \text{sim}(d, d')$ where $|\cdot|$ denotes the size of the set. Correspondingly, we define corpus homogenization as the average homogenization score of all essays. We use two metrics to compute the similarity between two documents: Rouge-L, an overlap-based metric; and BertScore (Zhang et al., 2020), an embedding-based metric.[5] Both homogenization metrics range from $0$ to $1$ with a higher score indicating more

---

[3] We use `gpt-3.5-turbo` due to its high performance on summarization on the HELM benchmark (Liang et al., 2022).

[4] See Table 6 in Appendix A.4 for the full example. On average, each essay is summarized into $6.86$ key points with a standard deviation of $1.53$. There is no significant difference between the average number of key points generated for essays from any pair of groups based on independent samples $t$-tests at the 5% level.

[5] We use the `microsoft/deberta-xlarge-mnli` model, the highest ranked model in the BertScore repository according to Pearson correlation with human ratings of similarity.

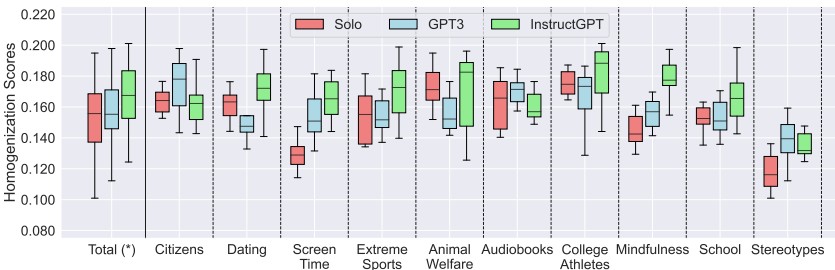

Figure 2: Boxplots of homogenization scores for all three groups (Solo, InstructGPT, GPT3) when comparing essays at the key point level using Rouge-L as the similarity metric. The left-most column (Total) shows essay homogenization scores for all topics and the other columns show the scores for essays on each topic. Essays written with InstructGPT exhibit higher corpus homogenization by a statistically significant margin (Section 4.2) and the highest median topic homogenization on 7 out of 10 topics. We observe the same trend when using BertScore in Figure 9(b) in Appendix C.

similar content. We measure homogenization at both the essay level and the key point level. For key point homogenization, we represent each essay as a concatenation of its key points (Section 3) before calculating its homogenization score.

## 4.2 RESULTS

**Writing with InstructGPT produces more similar content.** The corpus homogenization scores when comparing essays from the three groups at the key point level using Rouge-L are $0.1536$ for Solo, $0.1578$ for GPT3 and $0.1660$ for InstructGPT. InstructGPT exhibit higher corpus homogenization than both other groups to a statistically significant degree as verified with $p < 0.05$ on an independent samples $t$-test. This result holds across both similarity metrics as well as when the essays are compared at the essay level (Table 10 in Appendix C). Figure 2 shows the homogenization scores across topics at the key point level using Rouge-L.[6] We observe that the InstructGPT group has a higher median homogenization than other groups on 7 out of 10 topics. While the effect of model intervention varies across topics, the trend of increased homogenization persists (Appendix B.3).

**Writing with GPT3 does not increase homogenization.** Although GPT3 and InstructGPT contribute a similar amount of text and key points to the co-written essays (Section 3), unlike InstructGPT, GPT3 does not appear to increase the similarity between essays written by different users. In particular, there is no statistically significant difference between the corpus homogenization score of GPT3 and that of Solo at the 5% significance level using an independent samples $t$-test. This discrepancy between InstructGPT and GPT3 suggests that the impact of LLMs on content homogenization is not uniform. While InstructGPT often outperforms GPT3 on benchmarks across a wide range of metrics (Liang et al., 2022), our results highlight that this performance boost may come at the cost of creating more homogeneous content in interactive settings.

## 5 DOES WRITING WITH LLMS REDUCE THE OVERALL DIVERSITY?

In the previous section, we have seen that writing with the feedback-tuned model leads to different users writing similar essays on the same topic. A direct consequence of this homogenization effect is that it may reduce the overall diversity of the collection of writings from many authors. In this section, we test whether writing with LLMs reduce the diversity of the produced content.

### 5.1 METRIC

To measure the diversity of a collection of essays, we represent the corpus as a bag of information units (e.g., $n$-grams), and compute the number of unique information units divided by the total number of information units, such as the type-token ratio. Intuitively, if the essays all repeat each other (least diverse), the entire collection can be represented as a single essay (small ratio).

---

[6]Figure 8 and Figure 9 in Appendix C contain similar plots of the homogenization scores at the essay level and key point level using both Rouge-L and BertScore.

We consider two types of information units: $n$-grams and key points. The fraction of unique $n$-grams is a straightforward measure of lexical diversity (Li et al., 2023; Meister et al., 2023). For key point diversity, we first represent each essay as a set of key points (as described in Section 3) and aggregate the key points from all essays. To compute the number of unique key points, we perform agglomerative clustering and consider all key points in one cluster to be equivalent. We use the implementation of agglomerative clustering from Scikit-learn (Pedregosa et al., 2011) using the complete linkage criterion.[7] We convert Rouge-L to a distance metric for the clustering algorithm by subtracting the similarity score from 1 (which ranges from 0 to 1).

## 5.2 RESULTS

| $n$-gram size | Solo | GPT3 | InstructGPT |
|---|---|---|---|
| 1 | 0.119 | 0.116 | **0.115** |
| 2 | 0.602 | 0.585 | **0.579** |
| 3 | 0.898 | 0.886 | **0.869** |
| 4 | 0.973 | 0.967 | **0.953** |
| 5 | 0.991 | 0.988 | **0.977** |

(a) Fraction of unique $n$-grams

| Thresholds | Solo | GPT3 | InstructGPT |
|---|---|---|---|
| 0.5 | 0.982 | 0.971 | **0.950** |
| 0.6 | 0.941 | 0.927 | **0.877** |
| 0.7 | 0.792 | 0.779 | **0.738** |
| 0.8 | 0.543 | 0.514 | **0.494** |

(b) Fraction of unique key points.

Table 3: Content diversity calculated in two ways: (a) $N$-gram diversity in each setting. Bold values indicate the lowest diversity score as measured by the fraction of unique $n$-grams. Essays written with InstructGPT are the least diverse across $n$-gram sizes. (b) Diversity measured by agglomerative clustering of the key points of essays with RougeL using various distance thresholds for clustering. Bold values are different from other columns by a statistically significant margin ($p < 0.05$). InstructGPT consistently exhibits lower content diversity across all selected thresholds.

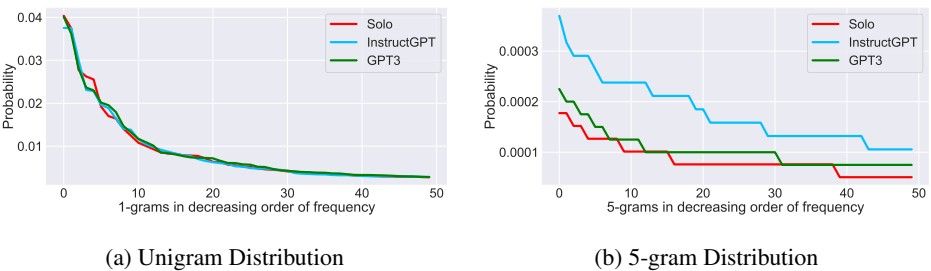

(a) Unigram Distribution      (b) 5-gram Distribution

Figure 3: Distributions of the top-50 (a) unigrams and (b) 5-grams from each group. Despite similar unigram distributions, essays written with InstructGPT exhibit higher repetition of common 5-ngrams.

**Writing with InstructGPT reduces lexical diversity.** In Table 3a, we show the diversity measured by $n$-grams (where $n = 1, 2, 3, 4, 5$) for essays written in all three groups. We see that the InstructGPT group consistently exhibits lower diversity across all values of $n$, whereas GPT3 does not significantly decrease the diversity score. The reduced lexical diversity showcases a disadvantage of co-writing with LLMs, particularly in creative writing genres like stories and poems, as it may wash away unique expressions on the long tail of human writings.

**Writing with InstructGPT reduces key point diversity.** In Table 3b, we report the diversity scores measured by the fraction of unique key points. The clustering process involves a threshold that determines the maximum distance above which two clusters will not be merged.[8] We vary this threshold and find that the InstructGPT group consistently exhibits lower diversity across all selected values.[9] In addition, the diversity of the InstructGPT group is lower than both the Solo and GPT3 groups by a statistically significant margin ($p < 0.05$) by a pairwise permutation test detailed in

---

[7]Complete linkage measures the distance between two clusters by the maximum distance between any pair of items from them.

[8]A higher distance threshold indicates that key points having lower similarity can be clustered together. At higher thresholds, the diversity becomes lower for all groups because more key points are considered equivalent.

[9]The same trend holds when using BertScore as a distance measure (Table 12b in Appendix C).

Appendix B.1. More concerning than reduced lexical diversity, this decline in key point diversity raises concerns with long-term social impact, emphasizing the need to safeguard perspectives from underrepresented groups as collaborative writing gains prominence.

**Essays written with InstructGPT repeat higher-order $n$-grams more frequently.** To further analyze how diversity is changed, we plot the distribution of unigrams and 5-grams in each group of essays in Figure 3 (truncated to the most frequent 50 $n$-grams from each group).[10] In the InstructGPT group, a notable concentration of probability mass in the head of the distribution for 5-grams suggests a higher degree of repetition and lower text diversity in the co-writing process. We verify the significance of the difference in 5-gram frequencies between InstructGPT and Solo via a chi-square test ($p < 0.05$).[11] To see what kind of phrases the model repeats, we list the ten most frequent 5-grams in Solo and InstructGPT in Table 4. We find that common 5-grams from InstructGPT often contain topical words overlapping with the essay prompt, whereas those from Solo are more often generic phrases. As users may seek suggestions for key ideas in the essays (Section 3), the similarity of the suggested content manifests in these common topical 5-grams which occur repetitively.

| Solo | | InstructGPT | |
|---|---|---|---|
| **5-Gram** | **Count** | **5-Gram** | **Count** |
| keeping up with the news | 7 | keep up with the news | 14 |
| in my opinion the most | 7 | on animal welfare when humans | 12 |
| keep up with the news | 6 | to focus on animal welfare | 11 |
| opinion the most important things | 6 | selfish to pursue risky sports | 11 |
| but on the other hand | 5 | students should learn in school | 11 |
| the most important thing that | 5 | wrong to focus on animal | 10 |
| wrong to focus on animal | 5 | sports like extreme mountain climbing | 10 |
| focus on animal welfare when | 5 | keeping up with the news | 9 |
| unfair when it is considered | 5 | the end of the day | 9 |
| in my opinion listening to | 4 | things students should learn in | 9 |

Table 4: Most frequent 5-grams in Solo and InstructGPT corpora along with the corresponding repetition counts. Highlighted 5-grams are those that contain topic-specific information from the prompts. We observe higher repetition overall and more occurrence of topical 5-grams with InstructGPT.

We additionally show in Appendix C that the reduction of diversity measured by linguistic units such as $n$-grams and key points also correlates with less diversity in an information-theoretic sense.

## 6 WHY DOES WRITING WITH INSTRUCTGPT REDUCE DIVERSITY?

We consistently observe that writing with InstructGPT results in a reduction of content diversity, whereas this is not the case with GPT3 despite users seemingly engaging equally with both models (Section 3). This raises the question of how writing with InstructGPT makes a difference. There are two potential contributors. First, InstructGPT may produce less diverse text, influencing the essay by directly contributing text. Second, users may write differently in the presence of assistance, e.g., adopting the model's opinion (Jakesch et al., 2023). In this section, we aim to answer this question by disentangling the effect from the model and the user in collaborative settings.

**InstructGPT generates less diverse text than GPT3.** We first confirm that generations from InstructGPT are less diverse than GPT3, which has also been observed in prior work. The technical report for GPT4 finds that the feedback-tuned model is less calibrated (OpenAI, 2023) and Bai et al. (2022) find that finetuning leads to decreased entropy of the output distribution. In our setting, we measure the diversity of model generation by the pairwise similarity of the five generated continuations upon each user query, similar to self-BLEU (Zhu et al., 2018). Thus, higher similarity scores indicate lower diversity. We observe that InstructGPT indeed generates less diverse text than GPT3 on average (0.20 vs 0.11 on Rouge-L), with significance at the 5% level using an independent samples $t$-test.[12] Since users incorporate text from both models at similar rates (Table 1), it is plausible that the less

---

[10]Each line in Figure 3 corresponds to the 50 most frequently repeated $n$-grams in a specific group, which vary across groups. Figure 10 in Appendix C shows similar plots for 1 to 5-grams.

[11]We observe a significant difference in $n$-gram usage for 4- and 5-grams as detailed in Appendix B.2.

[12]We also obtain the same conclusion using BertScore as the similarity metric (see Figure 12 in Appendix C).

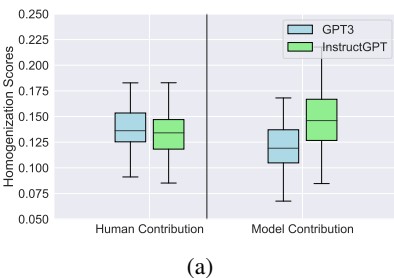 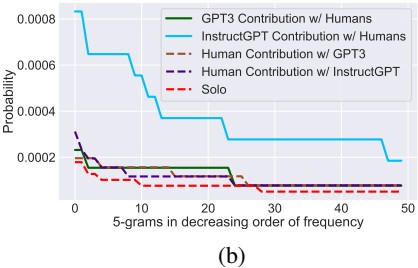

(a)                                                                                          (b)

Figure 4: (a) Homogenization scores calculated separately on key points attributed to users and the model using Rouge-L. InstructGPT contributed key points show a higher degree of homogenization whereas user-contributed key points have similar homogenization scores when writing with InstructGPT and GPT3. (b) Distributions of 5-grams introduced by the user (dashed lines) and the model (solid lines). The distribution of 5-grams from Solo essays is also provided as a reference. Distributions of user-written text in all settings are similar regardless of model assistance, whereas InstructGPT contributed text has notably larger probability mass on common 5-grams. This indicates that the reduced lexical diversity in the InstructGPT group is mainly due to model introduced text.

diverse suggestions from InstructGPT lead to a decrease in the diversity of the final co-written essay. Next, we examine the diversity of model-written and user-written text in the essays directly.

**InstructGPT increases repetition of higher-order $n$-grams while user-written text is unaffected.** The essays consist of both model-written and user-written text. Which contributes more to the decrease in diversity? To study this, we attribute each token in the essay to either the user or the model.[13] We then examine the distribution of 5-grams from user-written and model-written text Figure 4(b).[14] We see that the 5-gram distributions of user-written text remain the same regardless of whether the user writes with a model (Solo vs. GPT3/InstructGPT) and which model they write with (GPT3 vs. InstructGPT). Hence the phrase-usage pattern of users is not affected by the presence of model assistance. Instead, the increased repetition of 5-grams observed in Section 5.2 is more model-related as evidenced by the high probability mass on common 5-grams introduced by InstructGPT.[15]

**InstructGPT increases similarity between key points while user-written text is unaffected.** Finally, we disentangle the effect from the model and the user for the increase in homogenization observed in Figure 2. We attribute each key point to either the user or the model (same as in Section 3) and calculate the essay homogenization scores on user-contributed and model-contributed key points respectively. In Figure 4(a), we compare the user-contributed and model-contributed homogenization across the InstructGPT and GPT3 groups. We observe that, while the average homogenization score of user-contributed key points does not change between the two groups, the InstructGPT contributed key points have higher average homogenization than GPT3 ($p < 0.05$). This suggests that user behavior with respect to content creation is largely unchanged when writing with different models,[16] and the increased homogenization is mainly due to InstructGPT contributions.

**Takeaway.** Our results suggest that the users' lexical choice and argument generation is not affected when writing with model assistance. The reduction in diversity in collaborative writing is attributed more to text contributed directly by the InstructGPT model. This paints a relatively optimistic picture as the risk of forming a feedback loop through the mutual influence between the user and the model is potentially limited. However, it is important to note that our study focuses on single interactions between users and models, and the dynamics might change through repeated interactions over time.

---

[13]We use the recorded character-level keystroke information (Section 2.1) to determine the authorship of each token. Most tokens are attributed entirely to either the user or the model. In cases of mixed contribution, authorship is assigned based on the majority of characters.

[14]We ensure that each entire $n$-gram here was written by the user or model in a single continuous span of text to ensure we do not count incoherent $n$-grams.

[15]Figure 11 in Appendix C shows similar distributions varying $n$ from 1 to 5. We see the increased repetition of $n$-grams introduced by InstructGPT in 3 and 4-grams as well.

[16]We can also compare the homogenization of user-contributed key points with and without model assistance. However, we note that user-contributed key points in co-written essays are a subset of the original key points and hence exhibit more variation across essays. Therefore, they are not directly comparable to those in Solo.

## 7    RELATED WORK

**Human-AI collaborative writing.**    Collaborative writing predates LLMs where users are assisted by suggestions retrieved from a knowledge base (Swanson and Gordon, 2008; 2012; Chen et al., 2014; Roemmele and Gordon, 2015) or generated by task-specific RNNs (Ghazvininejad et al., 2017; Roemmele and Gordon, 2018; Clark et al., 2018). The early systems can produce relevant suggestions, but they often need to be revised before being incorporated into the draft. In contrast, LLMs offer suggestions that are often directly incorporated into user writing, either through continuations or in response to instructions, resulting in a surge of studies demonstrating their effectiveness (Gero and Chilton, 2019; Akoury et al., 2020; Yuan et al., 2022; Swanson et al., 2021; Chakrabarty et al., 2022; Ippolito et al., 2022; Mirowski et al., 2023) However, despite the improved productivity, we show that LLM-contributed text can also affect the writing in subtle and unintended ways.

**Effect of human-AI co-writing.**    Recent research focuses on analyzing user-model interactions and their influence on users and the writing process. Buschek et al. (2021) investigate how the number of suggestions affects users in email writing, highlighting the tradeoff between suggestion utility and writing efficiency. Lee et al. (2022a) collect data on user interactions with GPT3 as a collaborator, noting increased vocabulary diversity and productivity. Our work extends this research by formalizing metrics to quantify the homogenization effects resulting from user interactions with LLMs by comparing collaborative essays with essays written without model help.

**Evaluation of interactive text generation.**    Traditional evaluation metrics primarily assess text similarity to reference text (Papineni et al., 2002; Lin, 2004; Sellam et al., 2020). In interactive contexts where references are absent, alternative metrics gauge model assistance effectiveness indirectly. These include user-rated model helpfulness (Roemmele and Gordon, 2018; Clark et al., 2018; Padmakumar and He, 2022), retention of model suggestions in the final text (Akoury et al., 2020), and completion time (Buschek et al., 2021). Lee et al. (2022b) additionally considers user ownership and enjoyment which emphasizes maintaining user writing style. Our work extends existing metrics that evaluate lexical diversity and text style based on token and POS n-grams statistics (Roemmele et al., 2017; See et al., 2019; Tevet and Berant, 2021; Meister et al., 2023). We adapt these metrics for co-writing scenarios and introduce content-specific diversity measures.

**Social impacts of LLMs.**    Bommasani et al. (2021); Bender et al. (2021); Solaiman et al. (2023) extensively discuss the societal risks of the adoption of LLMs in user-facing applications. Most relevant to our work is the concern over the potential emergence of an algorithmic monoculture (Kleinberg and Raghavan, 2021), which assesses outcome homogenization in decision-making Creel and Hellman (2022); Bommasani et al. (2022). Additionally, Hancock et al. (2020) discuss the potential language homogenization during AI-mediated communication, while Arnold et al. (2020) demonstrate that predictive keyboards lead to shorter image captions with fewer rare words. Jakesch et al. (2023); Bhat et al. (2023) show that writing with a biased language model can change user opinions. Liao and Xiao (2023) advocate for novel evaluation methods to uncover socio-technical issues arising from increased general-purpose LLM use. Our work investigates one axis of such evaluation, namely content diversity, in co-writing settings.

## 8    CONCLUSION

In this work, we study how writing with LLMs impacts the diversity of content produced. Through a controlled study, we find that users writing with InstructGPT produce more similar content than those writing with GPT3 or without model help. This homogenization also results in a reduction of the overall diversity of content produced by many users. Furthermore, our analysis indicates that this effect is attributable to the less diverse text contributed by InstructGPT, while the user-contributed text remains largely consistent with and without model help. These findings highlight a new axis on which to evaluate the impact of LLMs prior to their deployment. While adapting a model with human feedback leads to an improvement in instruction following, this may be accompanied by a reduction in content diversity. Combined with other recent findings that LLMs influence user opinions during the writing process (Jakesch et al., 2023), this calls for a careful user-centered evaluation to ensure that these models do not suppress the voice of users in scenarios where personal expression is desired. Finally, our work also adds to the growing body of open problems on reinforcement learning with human feedback (Casper et al., 2023). Learning from the diverse feedback of many users and personalizing model generations to individuals are challenging future directions.

ACKNOWLEDGEMENTS

We would like to thank Mina Lee, Nick Lourie, Naomi Saphra, Richard Pang, and Nitish Joshi for their input at various stages of the project. We would also like to thank Abby Rabinowitz and Alexander Landfair from the NYU Expository Writing Program for valuable discussions during the planning stages. We would finally like to acknowledge all the writers recruited for this project without whom this work would not have been possible. This work is supported by Open Philanthropy, AWS AI, and the National Science Foundation under Grant No. 1922658.

REPRODUCIBILITY STATEMENT

All data collection experiments were done between March and July 2023. For the sake of reproducibility, the essays with character-level logs as recorded by the interface along with all model suggestions presented to users will be released after the review period. This allows readers to replay entire writing sessions using the code released by Lee et al. (2022a). We also make public all the scripts to process the raw logs to perform the analyses run in this paper.

ETHICS STATEMENT

We detail all the procedures for user recruitment and remuneration in Appendix A.1. The plan for our user study was approved by the Institutional Review Board of our university. We obtained consent from all participants to share the essays publicly post the completion of the study. We detail further limitations of our study in Appendix D.

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

## A  TASK DETAILS

### A.1  USER RECRUITMENT ON UPWORK

We recruit a total of 38 participants from Upwork, all of whom are native English speakers and have prior experience in writing or copyediting on the platform. Their prior experience varied from 5 to over 200 previously completed tasks on the platform. Our participants are exclusively from the United States and encompassing diverse racial backgrounds with an approximately equal distribution between genders. Each participant is asked to complete a session consisting of three essay writing tasks—one in each of the three settings and each on a different topic. In each session, the order of the three settings and the assignment of topics are randomized. We don't enforce that users include a certain amount of machine-generated text as we want to study how collaborative writing in its most natural form differs from solo writing without model help. The average machine-written fraction is 0.32 for InstructGPT and 0.35 for GPT (Table 9) indicating that users obtain useful help from the models without over-relying on the same. The vast majority of essays have machine-written fractions ranging from 0.2 to 0.5 with a few outliers being as low as 0.1 and as high as 0.8 (Figure 13 in Appendix C). Users are allowed to edit their essays post-completion. Their objective is to convey their opinions effectively. We acknowledge that this might affect word usage patterns which again motivates the contribution of our proposed key point-based evaluation as these key points are less likely to be changed during post-editing. Each essay takes 15 minutes on average and hence a session usually takes the participants under one hour. Compensation was provided through hourly contracts, prorated to $20 per hour. After each participant completed one session, their essays were reviewed manually by the authors of this work for the relevance to the essay prompt as well as coherence of writing. This process does not involve policing the actual content of the essays and we encourage our participants to express their opinions in detail. This was used to filter out a few participants whose sessions were reassigned. The remaining participants were then invited to complete two additional sessions, each with different topics. As a result, each of the 38 participants completed between 1 and 3 sessions. In total, we obtain 10 essays on each of the 10 topics for each setting, resulting in 100 essays per setting.

### A.2  DECODING PARAMETERS ON OPENAI API

To avoid reduced diversity due to decoding, we use a decoding strategy biased towards higher diversity. Specifically, we sample continuations from both models following the "high randomness" setting from Lee et al. (2022a). The decoding parameters we use are:

- Engine: `davinci` for the base language model and `text-davinci-003` for the feedback-tuned language model
- Response length (word piece): 30
- Temperature: 0.9
- Top P: 1
- Frequency penalty: 0.5
- Presence penalty: 0.5

| Prompt Code | Prompt (Source URL) |
|---|---|
| school | What Are the Most Important Things Students Should Learn in School? In your opinion, what are the most important things students should learn in school? What is the most important thing you have learned in school? How has this knowledge affected your life? How do you think it will help your success in the future? (Link) |
| stereotype | What Stereotypical Characters Make You Cringe? What stereotypical characters in books, movies or television shows make you cringe and why? Would you ever not watch or read something because of its offensive portrayal of someone? (Link) |
| audiobook | Is Listening to a Book Just as Good as Reading It? Do you listen to audiobooks? What are the benefits, in your opinion, of listening instead of reading? Are there advantages to reading that cannot be gained by listening? Which method do you prefer? Why? (Link) |
| athletes | Should College Athletes Be Paid? Do you think college athletes should be paid? Or is a college scholarship and other non-monetary perks like the opportunity to play in front of cheering fans enough? What possible difficulties or downsides might there be in providing monetary compensation to players? (Link) |
| extremesports | Is It Selfish to Pursue Risky Sports Like Extreme Mountain Climbing? Some sports, like extreme mountain climbing, are dangerous. Since there are varying degrees of risk in most, if not all, sports (such as the possibility of concussions, broken bones and even death), how does one decide where the line might be drawn between what is reasonable and what is not? Are some sports simply too dangerous to be called a sport? (Link) |
| animal | Is It Wrong to Focus on Animal Welfare When Humans Are Suffering? Would you be surprised to hear that a study found that research subjects were more upset by stories of a dog beaten by a baseball bat than of an adult similarly beaten? Or that other researchers found that if forced to choose, 40 percent of people would save their pet dog over a foreign tourist. Why do you think many people are more empathetic toward the suffering of animals than that of people? In your opinion, is it wrong to focus on animal welfare when humans are suffering? Why do you think so? (Link) |
| news | Are We Being Bad Citizens If We Don't Keep Up With the News? In your opinion, are we being bad citizens if we don't keep up with the news? Do you think all people have some responsibility to know what is going on in the world? Does engaging with current events actually do anything at all? Why do you think the way you do? (Link) |
| mindfulness | Should Schools Teach Mindfulness? Have you ever tried mindfulness or meditation, practices that focus on the present moment and being aware of your thoughts, feelings and bodily sensations? If so, what was it like for you? If not, does it sound like something you'd like to try? Do you think that such practices have a place in schools? Why or why not? (Link) |
| screen | How Worried Should We Be About Screen Time During the Pandemic? The coronavirus pandemic ended the screen time debate: Screens won. We all now find ourselves on our screens for school, for work and for connecting with family and friends during this time of social distancing and increased isolation. But should we be worried about this excessive screen use right now? Or should we finally get over it and embrace the benefits of our digital devices? (Link) |
| dating | How Do You Think Technology Affects Dating? Have you had any experience with dating? Have you ever used dating apps? If so, what has it been like for you? If not, why not? How do you think technology — like apps, Netflix, social media and texting — affects dating and relationships? In your opinion, does it improve or worsen romantic interactions? (Link) |

Table 5: List of prompts used in the experiments.

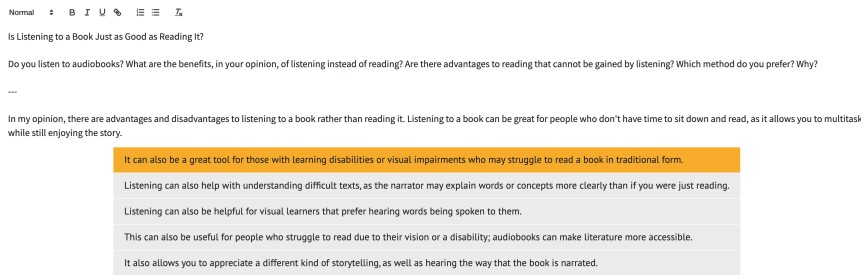

Figure 5: Text editor interface showing suggestions when requested by the user.

## A.3 USER INSTRUCTIONS

**Essay Requirements**  To complete an essay assignment successfully, you would need to write a short 3-passage essay given the prompt presented to you. The essay should reflect your opinion of the assigned topic and you're writing an argumentative piece for why you feel that way. The expected length of the essays is around 300 words (3-4 short paragraphs of 3-4 sentences each). Each piece should take you between 10 and 15 minutes (as calculated from prior experiments). When writing with AI help, you are required to obtain model suggestions at least 5 times (we encourage you to do this more if you find it helpful, more the better). Some example responses are provided below (though these were written by non-experts so I'm sure you can identify flaws and improve on the style)

**Instructions in Detail**

- When you are completing a task, make sure you have a Session ID. If you don't have this already, please send us a message on Upwork.

- View the spreadsheet of assignments and navigate to your assigned session. Each session will correspond to three rows in the spreadsheet, each with an accompanying URL.

- Each of these links corresponds to the three assignment essays you would have to complete in order to finish the task.

- If you click one of these, you will see the text editor pop up which looks like this which shows the prompt for the assigned essay.

- Out of the three assignments, please note that one is the essay that you will have to write without the assistance of the AI. Please complete the three assignments in the order provided.

- When you write with the AI, you will have the option to hit the TAB key on your keyboard and obtain suggestions from the model in the form of a dropdown like this. You are free to continue the writing process as you normally would, however, we encourage you to make use of model help when you are looking for ideas when you are stuck or just want to take a look at some possible continuations :)

- When you write with the AI, you need to request at least 5 suggestions from the model. You do NOT need to accept all of the suggestions, you are free to accept one then edit it, ask for suggestions again at the same time, or even just reject all of the model suggestions. We encourage you to use the model as much as possible :) The objective of the study is to understand the effect of the model, so more interaction is better.

- Once you complete the essay, make sure you hit 'Save' at the bottom of the screen to obtain a verification code on completion. We ask that you complete the writing in one sitting. We aim to record the entire writing process so please refrain from copy-pasting any text into the editor.

---

**Essay**

While I believe the concerns regarding children's screen time are valid, I believe it is somewhat biased to not take this problem, which is a genuine issue right now, as an everyone problem, and not a student one. With that in mind, I do believe that for the majority of our generation it would not be as hard to disconnect with devices after the pandemic.

I know that I, along with many other teenagers, would like nothing more than to go back to school, play sports outside, meet new people and such. I doubt I could say the same for our older generations, because while parents ban teenagers from using their phones, they often spend many more hours on their devices. Cranky grandparents grumble about how 'back in their day they lived without a mobile phone' usually forget the point of their stories. Back in their day indeed.

We live in a different era, and to think that the same rules should apply to teenagers, especially during a pandemic, isn't logical.Pinterest, Youtube, Twitter aren't just places where teens while away their time. They are also places for new ideas, watching college lectures, and political discourse. Limiting screen time, while in the short run is helpful, is doubtful to do anything in the long run. Banning technology is not stopping kids from using it. It's just stopping them from telling their parents about it.

I would like to end by saying, parents, please trust your teenager just a little bit more, and don't be too worried about their screen time!

**Keypoints**

- The concerns regarding children's screen time are valid
- The problem of screen time should be considered an everyone problem, not just a student one
- For most young people, it would not be hard to disconnect from devices after the pandemic
- Older generations often spend more time on their devices than teens, despite their complaints
- Social media can be used for educational and constructive purposes
- Limiting screen time may not be effective in the long run
- Parents should trust their teenager more and not worry too much about their screen time

---

Table 6: Example of summarization into keypoints

### A.4 SUMMARIZATION INTO KEY POINTS

Motivated by recent encouraging results of using LLMs for zero-shot summarization (Goyal et al., 2022), we summarize each essay into a list of key points by prompting `gpt-3.5-turbo`. Table 6 contains a full example of an essay with the generated key points. We use a simple prompt, "Summarize this essay into a set of simple and distinct bullet points. Make sure that the bullet points cover all the information from the essay."

## B SIGNIFICANCE TESTING

### B.1 PERMUTATION TEST FOR SIGNIFICANCE OF DIVERSITY RESULTS

To test for the significance of the results on diversity (Section 5.2), since we do not have multiple instances of essay corpora from each setting (Solo, GPT3 and InstructGPT), we employ three permutation tests between each pair of settings. A walkthrough example of the permutation test setup is as follows. We wish to evaluate the significance of the difference between the diversity scores of Solo and InstructGPT, as measured by the clustering of key points (Table 3) or by traditional lossless

compression algorithms (Table 13). We first calculate the statistic (difference between the diversity scores) on the collected corpora of essays. We then take the union of Solo and InstructGPT essays, randomly partition this into two equal sets, and recalculate the statistic on these two permuted sets of essays. This process is then repeated for 1000 different permutations. We then obtain the p-value of this two-tailed permutation test as the proportion of times the absolute value of the statistic calculated on the permuted data is greater than the statistic calculated on the observed data from the user study. This is then repeated for all pairs out of Solo, GPT3, and InstructGPT. Bold values in the results tables (Table 3, Table 13) indicate those instances where the p-value on the permutation test for a setup (InstructGPT) was significant at the 5% level over both other setups (GPT3 and Solo).

### B.2 Chi-Square Test for Significance for $n$-gram Distributions

To confirm the significance of the difference between the categorical distributions of $n$-grams in the different setups, we employ a chi-square test on the count of occurrences. We evaluate if writing with InstructGPT results in a change of 5-gram usage in Figure 3 and Figure 4(b). To perform this test, we first identify and take the union of the 50 most frequently occurring 5-grams in each setup. We then obtain the counts of occurrences of each of these 5-grams in both corpora and perform a chi-square test for significance on these frequencies. A p-value $< 0.05$ results in a rejection of the null hypothesis and the observation of a significant difference in the categorical distributions. We perform these for all pairs out of Solo, GPT3, and InstructGPT and find that the repetition of $n$-grams when writing with InstructGPT is more similar by a significant margin compared to both other setups (Figure 3).

### B.3 Significance Testing for Results on Homogenization

As noted in Section 2.2 and Appendix A.1, we recruit a fixed set of users who wrote essays in each of the three setups in a randomized order. We ensure that they write essays on different topics in each setup to prevent a repetition of their opinions. To test for the significance of the change in homogenization across all topics between the three setups, we conducted independent samples t-tests in Figure 2. To further test the significance of this result, we observe that within each topic, each writer only submits an essay to one of the three setups. This matches the between-group experimental setup, albeit with less power as we only collected 10 essays per setup per topic. We compute the significance of the difference in homogenization scores per topic via an independent samples t-test. We report the p-values comparing each pair of setups within each topic at both the key point (Table 7) and essay level (Table 8). Comparisons with significance at the 5% level are marked with an asterisk. We observe that the InstructGPT setup exhibits higher homogenization over Solo writers with significance at the 5% level on 8 out of 10 topics at the raw essay level and 6 out of 10 at the key point level using both Rouge-L and BertScore. The same trend holds on GPT3 on 7 topics at the raw essay level and 5 at the key point level. We believe that this confirms the overall trend that writing with InstructGPT results in higher homogenization in the essays created.

## C Additional Results

**Reporting more basic statistics about the collected essays**     We aim to improve the comprehensiveness of our analysis by reporting additional metrics on the collected essays in Table 9.

- We compute basic statistics such as perplexity (via GPT2), average sentence length, and essay length on the essays collected. Predictably, writing with model-help reduces the perplexity of the essays which now incorporate suggestions sampled from an LM distribution. The total length of essays, in terms of word count, is roughly similar, with writers writing slightly shorter sentences when writing with InstructGPT (albeit with a slightly high standard error).

- Additionally, by calculating the average height of the dependency parse trees of the sentences in the essays, we observe that writing with InstructGPT results in sentences with fewer nested structures, indicative of slightly lower complexity.

- Writing with model help also results in fewer unique POS-Ngrams again providing some evidence that collaborative writing could result in more homogenized usage of language. The effect here is even between GPT3 and InstructGPT, a slight departure from the results

| Significance Testing - Key point level | | | | | | |
|---|---|---|---|---|---|---|
| **Rouge-L** | | | | **BertScore** | | |
| Topic | Solo vs In-structGPT | InstructGPT vs GPT3 | GPT3 vs Solo | Topic | Solo vs In-structGPT | InstructGPT vs GPT3 | GPT3 vs Solo |
| **Stereotype** | 0.02* | 0.34 | 0.01* | **Stereotype** | 0.13 | 0.18 | 0.81 |
| **School** | 0.03* | 0.05* | 0.86 | **School** | 0.78 | 0.15 | 0.17 |
| **Mindfulness** | 0.00* | 0.00* | 0.03* | **Mindfulness** | 0.00* | 0.11 | 0.64 |
| **Athletes** | 0.24 | 0.14 | 0.53 | **Athletes** | 0.09 | 0.05* | 0.67 |
| **Audiobook** | 0.73 | 0.03* | 0.25 | **Audiobook** | 0.00* | 0.06 | 0.01* |
| **Animal Welfare** | 0.67 | 0.13 | 0.01* | **Animal Welfare** | 0.31 | 0.02* | 0.00* |
| **Extreme Sports** | 0.03* | 0.02* | 0.93 | **Extreme Sports** | 0.43 | 0.11 | 0.38 |
| **Screen Time** | 0.00* | 0.09 | 0.00* | **Screen Time** | 0.00* | 0.00* | 0.06* |
| **Dating** | 0.17* | 0.00* | 0.04* | **Dating** | 0.01* | 0.01* | 0.41 |
| **Citizens** | 0.51 | 0.08 | 0.16 | **Citizens** | 0.16 | 0.11 | 0.87 |

Table 7: Evaluating the significance of the difference of homogenization between the three setups across each topic (Corresponding to the box plots in Figure 9). We conduct an independent samples t-test to compare the document homogenization values between the setups pairwise. Pairs with a significant difference at the 5% level are highlighted with an asterisk. The overall trend is that InstructGPT has significantly higher homogenization across a majority of topics.

| Significance Testing - Raw essay level | | | | | | |
|---|---|---|---|---|---|---|
| **Rouge-L** | | | | **BertScore** | | |
| Topic | Solo vs In-structGPT | InstructGPT vs GPT3 | GPT3 vs Solo | Topic | Solo vs In-structGPT | InstructGPT vs GPT3 | GPT3 vs Solo |
| **Stereotype** | 0.02* | 0.01* | 0.00* | **Stereotype** | 0.02* | 0.01* | 0.03* |
| **School** | 0.17 | 0.12 | 0.71 | **School** | 0.14 | 0.22 | 0.62 |
| **Mindfulness** | 0.00* | 0.01* | 0.42 | **Mindfulness** | 0.02* | 0.22 | 0.81 |
| **Athletes** | 0.00* | 0.00* | 0.52 | **Athletes** | 0.00* | 0.00* | 0.90 |
| **Audiobook** | 0.01* | 0.15 | 0.69 | **Audiobook** | 0.00* | 0.00* | 0.76 |
| **Animal Welfare** | 0.49 | 0.11 | 0.15 | **Animal Welfare** | 0.13 | 0.00* | 0.00* |
| **Extreme Sports** | 0.35 | 0.00* | 0.00* | **Extreme Sports** | 0.04* | 0.00* | 0.09 |
| **Screen Time** | 0.00* | 0.00* | 0.00* | **Screen Time** | 0.00* | 0.00* | 0.70 |
| **Dating** | 0.00* | 0.61 | 0.00* | **Dating** | 0.88 | 0.01* | 0.02* |
| **Citizens** | 0.00* | 0.14 | 0.11 | **Citizens** | 0.00* | 0.04* | 0.11 |

Table 8: Evaluating the significance of the difference of homogenization between the three setups across each topic (Corresponding to the box plots in Figure 8). We conduct an independent samples t-test to compare the document homogenization values between the setups pairwise. Pairs with a significant difference at the 5% level are highlighted with an asterisk. The overall trend is that InstructGPT has significantly higher homogenization across a majority of topics.

| | | Solo | GPT3 | InstructGPT |
|---|---|---|---|---|
| **Perplexity of Essays (via GPT2)** | | 25.067 | 22.10 | **20.26** |
| **Sentence Length (in words)** | | 23.51 (0.25) | 23.66 (0.24) | 22.51 (0.24) |
| **Height of Syntax Tree** | | 5.93 (0.06) | 5.98 (0.06) | 5.71 (0.05) |
| **Essay Length (in words)** | | 376.44 (8.03) | 380.87 (9.30) | 368.39 (8.43) |
| **Human Ratings** | **Relevance to Prompt** | 4.30 | 4.15 | 4.10 |
| | **Grammaticality** | 4.00 | 4.10 | 4.00 |
| | **Depth of Discussion** | 3.95 | 3.85 | 3.80 |
| **Unique POS Ngrams** | **1** | 73 | 56 | **58** |
| | **2** | 487 | 437 | **426** |
| | **3** | 1297 | 1261 | **1235** |
| | **4** | 2044 | **1975** | 1988 |
| | **5** | 2423 | **2338** | 2414 |

Table 9: Reporting basic statistics about the collected essays. Writing with the help of a model reduces the perplexity (measured via GPT2) of the essays in InstructGPT and GPT3. We also observe that users write essays of similar length across all setups with slightly shorter sentences in InstructGPT. We also examine the height of the dependency parse trees of the sentences in each setup to find that writing with InstructGPT also results in sentences with less complexity on average. Along with these average numbers, we report the standard errors in brackets. We also find that writing with model help results in fewer unique POS-$n$-grams. Finally, we rate the quality of one essay per topic per setup (selected randomly) along three different dimensions (grammaticality, relevance to the prompt, and depth of discussion) to find that there is no significance in essay quality across setups.

> on diversity which is more focused on content. We finally plot the distributions of the 50 most frequent POS-Ngrams (Figure 6) and find that on 4 and 5-grams, writing with model help leads to higher repetition of frequent POS-Ngrams.

- To ensure that the quality of the essays is not a confounder in the result, we present human ratings for a random sample of 10 essays (one per topic) from each setup that we conducted as part of our quality control checks. Here we judged the essays on 3 different axes—relevance to the topic, grammaticality, and the depth of discussion on their opinions. This was per the advice from faculty at the Expository Writing program at our university. This analysis showed no clear differences between the setups, explained in part due to the observation that the machine-written fraction in the collaborative essays was on average around 32%-35% (Table 9).

**Writing with InstructGPT results in more similar content at the key point and raw essay level** In addition to the results presented in Section 4.2, we also report the corpus homogenization scores at the key point and essay level in Table 10. Across both Rouge-L and BertScore, the corpus of essays written with InstructGPT exhibits higher corpus homogenization than both other groups by a statistically significant margin (p-value $< 0.05$) on both levels. We also plot the homogenization scores of all the essays calculated at the raw essay and key point levels in Figure 8 and Figure 9. We choose to report the homogenization at the key point level mainly because string similarity metrics such as Rouge-L and BertScore are less reliable on longer text documents (Sun et al., 2019; Gehrmann et al., 2023).

**Essays written with InstructGPT repeat higher- order n-grams more frequently** Building on from the results in Section 5.2, we plot the $n$-gram distributions of the raw essays varying $n$ from 1 to 5 in Figure 10. Here we truncate the distribution to the most common 50 $n$-grams from each setup. While the distributions for 1, 2 and 3-grams are almost identical across the setups, on 4 and 5-grams we see a concentration of probability mass at the head of the distribution for the essays written with InstructGPT. The reduction in lexical diversity (Table 3a) is manifested in this increased repetition of higher-order $n$-grams.

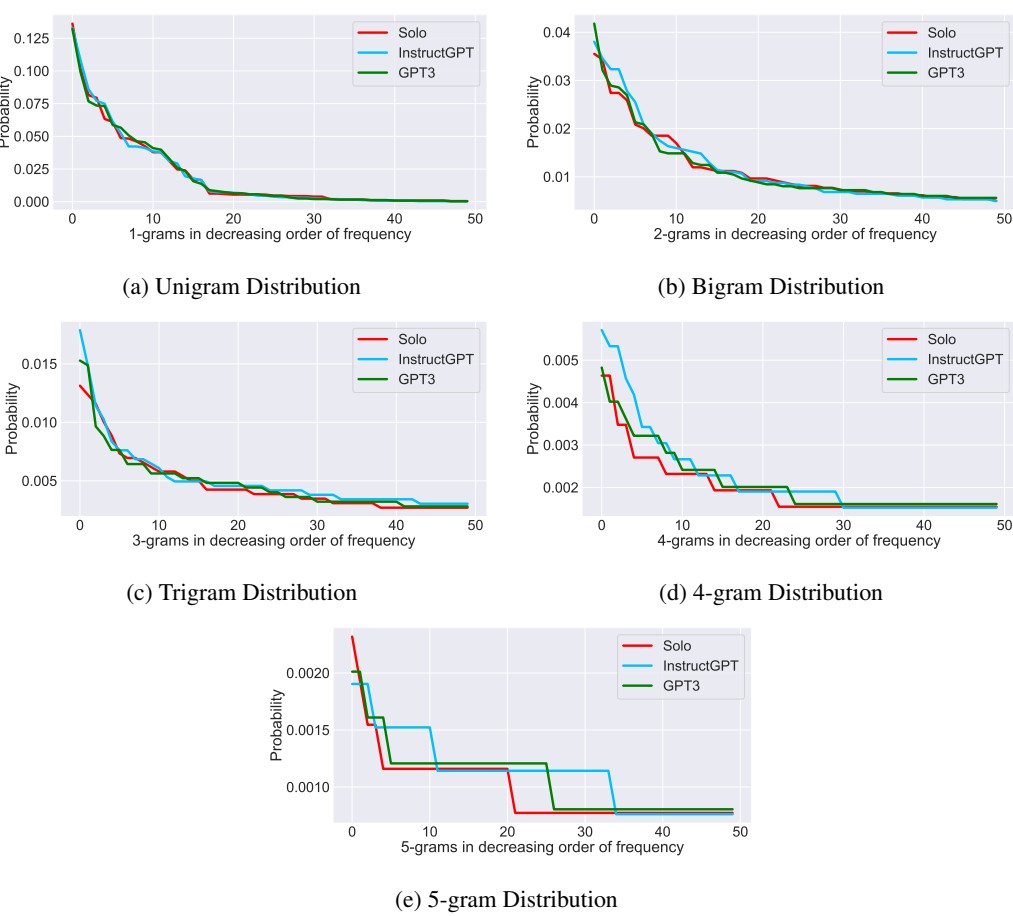

(a) Unigram Distribution

(b) Bigram Distribution

(c) Trigram Distribution

(d) 4-gram Distribution

(e) 5-gram Distribution

Figure 6: Distribution of the top-50 POS-$n$-grams in the essays from the various setups varying $n$ from 1 to 5. While the lower order $n$-gram usage patterns are similar across groups, essays written with InstructGPT exhibit higher repetition of common 4- and 5-POS-ngrams.

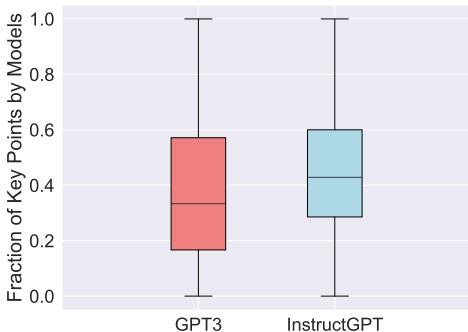

Figure 7: Boxplot of the fraction of key points contributed by the model in each essay. Both models contribute a considerable amount of key points. However, the high variance suggests varied reliance on the model by different users.

|  |  | Solo | GPT3 | Instruct-GPT |
|---|---|---|---|---|
| **Key point** | **Rouge-L** | 0.1536 | 0.1578 | **0.1660** |
| **Level** | **BertScore** | 0.6305 | 0.6284 | **0.6437** |
| **Essay** | **Rouge-L** | 0.1498 | 0.1523 | **0.1621** |
| **Level** | **BertScore** | 0.6044 | 0.6130 | **0.6318** |

Table 10: Corpus homogenization scores comparing essays from the three setups at the key point and essay level using both Rouge-L and BertScore to calculate homogenization. Essays written with InstructGPT exhibit higher homogenization than Solo and GPT3 across both levels and similarity metrics by a statistically significant margin (p-value $< 0.05$).

**Essays written with InstructGPT are more compressible.** We also show that the reduction of diversity measured by linguistic units such as $n$-grams and key points also correlates with less diversity in an information-theoretic sense. Specifically, we measure the compression ratio of essays written in the three settings using various lossless compression algorithms: LZMA, ZLIB, and GZIP, each being a variation of the Lempel-Ziv algorithm (Ziv and Lempel, 1977; 1978). These algorithms compress data by identifying repeated patterns and using dictionary mappings for encoding. To compute the compression ratio, we concatenate all essays from a setting into a single text file. We then run the compression algorithm on it and calculate the ratio of the compressed file size to the original file size. From Table 13, we see that the InstructGPT essays are consistently more compressible across all three compression algorithms, which is statistically significant with $p < 0.05$ according to permutation tests (Appendix B.1).

**InstructGPT presents less diverse suggestions to users than GPT3** To test the hypothesis that adapting a model with human feedback reduces the diversity of the presented suggestions, we calculate the average similarity of all pairs of the five suggestions presented by InstructGPT and GPT3 upon each user query. In addition to the similarity plotted with Rouge-L in Figure 12 we also plot the average similarity scores of suggestions computed using BertScore in Figure 12(b).

**Higher correlation between AI written fraction of the document and homogenization, on InstructGPT than GPT3** To study the relationship between increased model intervention and homogenization, we plot the fraction of the essay written by the model as a function of the document homogenization with BertScore and Rouge-L in Figure 13. We observe a weak correlation when users write with InstructGPT, particularly in the case of BertScore.

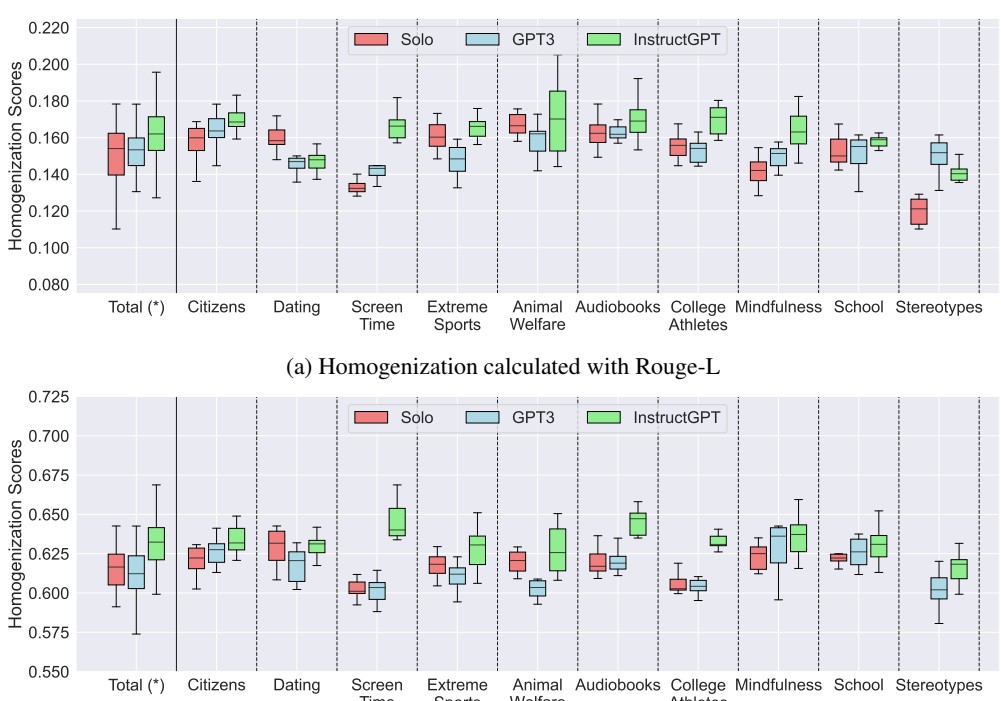

(a) Homogenization calculated with Rouge-L

(b) Homogenization calculated with BertScore

Figure 8: Boxplots of homogenization scores for all three groups (Solo, InstructGPT, GPT3) when comparing the raw essays and calculated using (a) Rouge-L and (b) BertScore as measures of similarity. The left-most column (Total) shows essay homogenization scores for all topics and the other columns show essay homogenization scores by topic. Essays written with InstructGPT exhibit higher corpus homogenization by a statistically significant margin (Section 4.2).

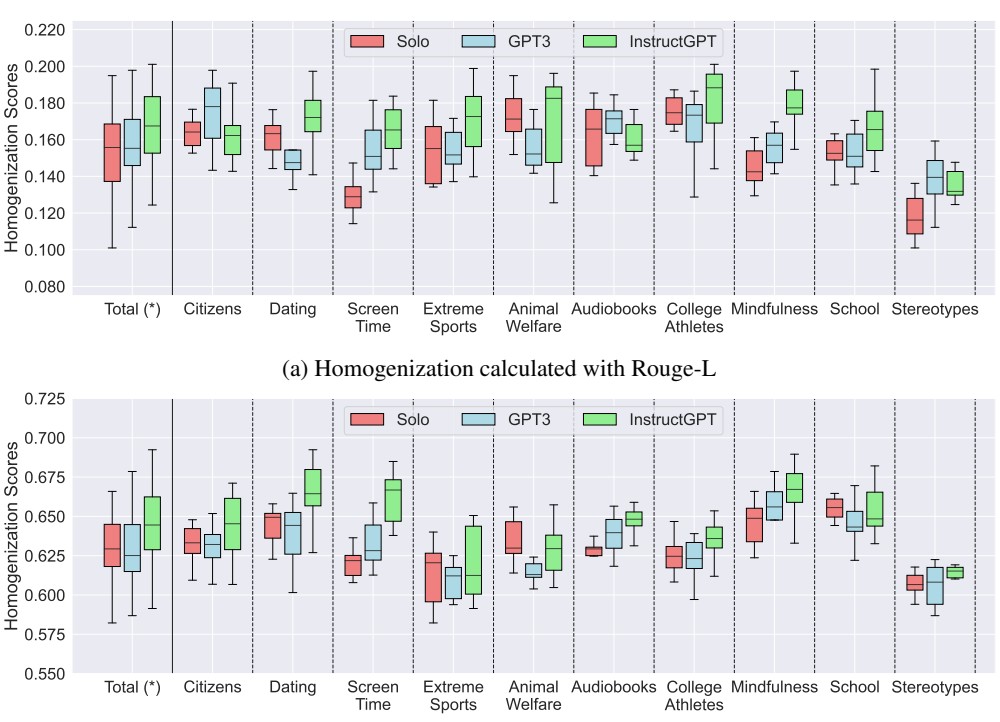

(a) Homogenization calculated with Rouge-L

(b) Homogenization calculated with BertScore

Figure 9: Boxplots of homogenization scores for all three groups (Solo, InstructGPT, GPT3) when comparing essays at the key point level and calculated using (a) Rouge-L and (b) BertScore as measures of similarity. The left-most column (Total) shows essay homogenization scores for all topics and the other columns show essay homogenization scores by topic. Essays written with InstructGPT exhibit higher corpus homogenization by a statistically significant margin (Section 4.2).

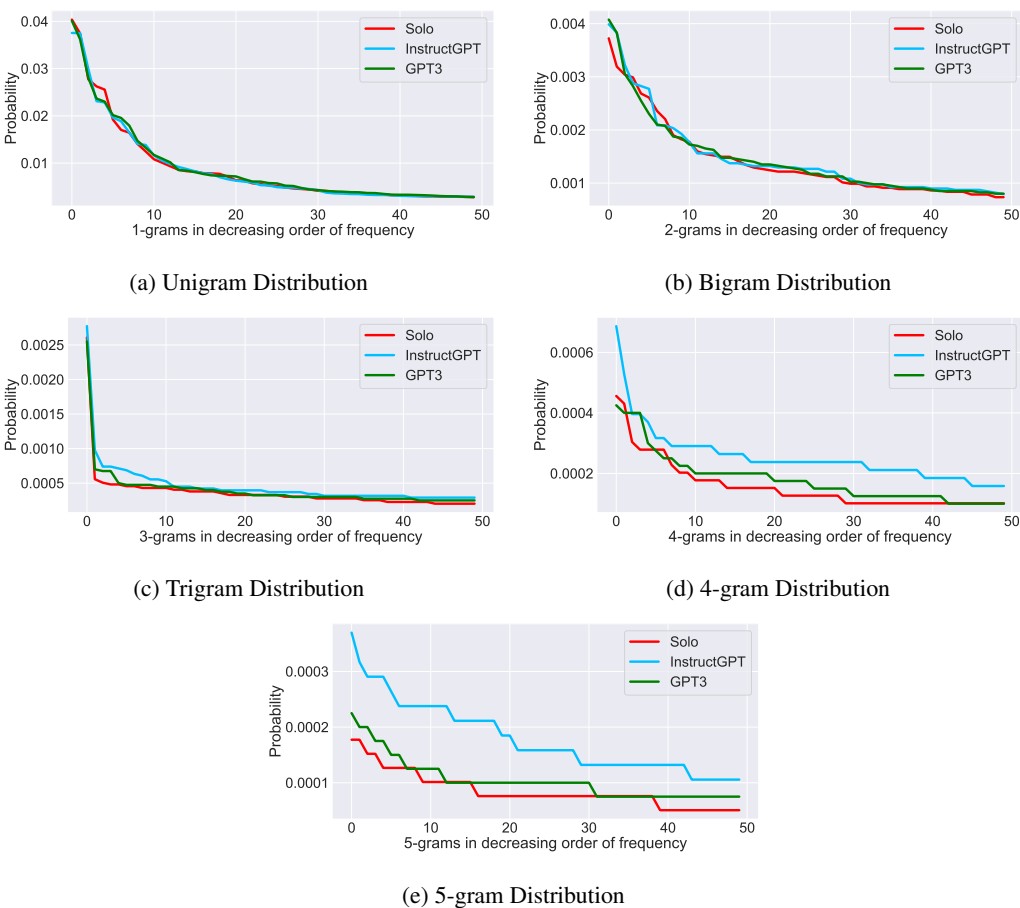

Figure 10: Distribution of the top-50 $n$-grams in the essays from the various setups varying $n$ from 1 to 5. While the lower order $n$-gram usage patterns are similar across groups, essays written with InstructGPT exhibit higher repetition of common 4-ngrams and 5-grams.

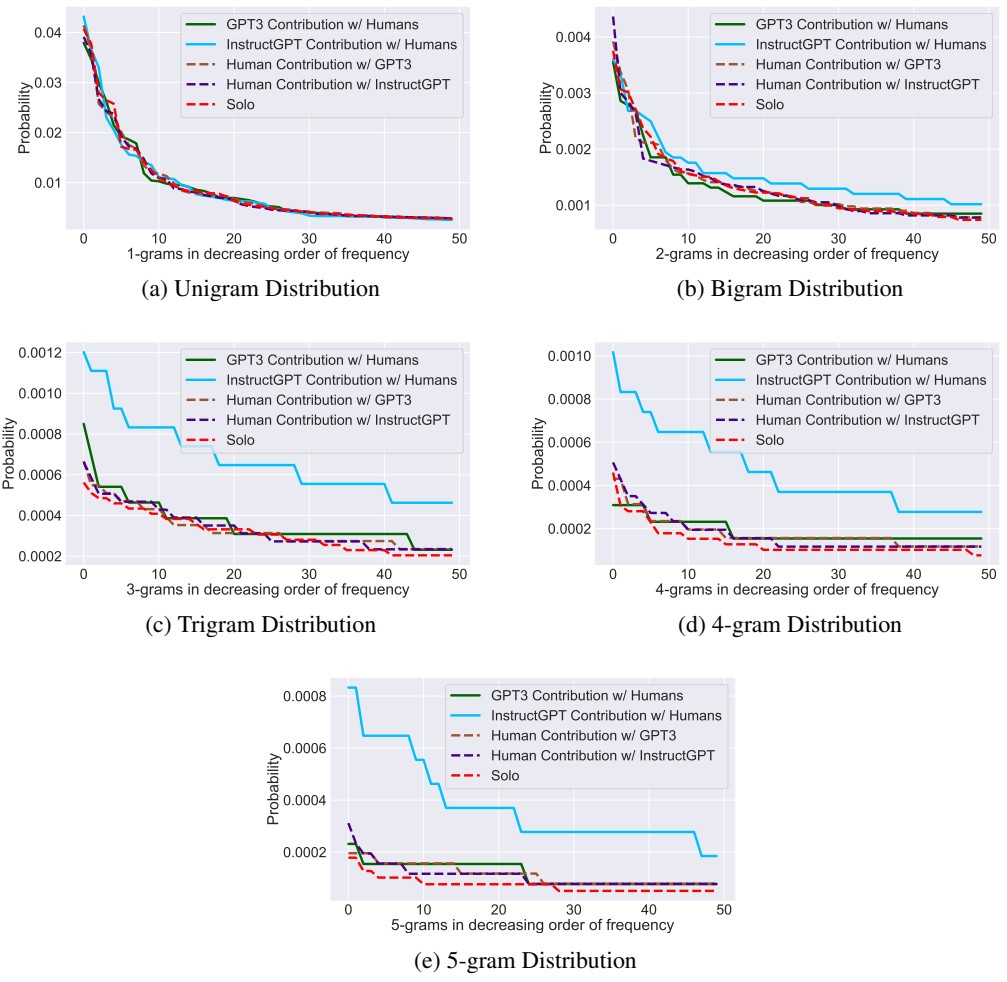

(a) Unigram Distribution      (b) Bigram Distribution

(c) Trigram Distribution      (d) 4-gram Distribution

(e) 5-gram Distribution

Figure 11: $n$-gram distribution of tokens introduced by the user and the model during the two co-writing setup with InstructGPT and GPT3 respectively. The distribution of human-written $n$-grams from Solo essays is also provided as a reference. The distribution of user-written text in all settings is similar to each other regardless of model assistance, whereas InstructGPT contributed text notably has larger probability mass on common $3$, $4$, and $5$-grams. This indicates that the reduced lexical diversity in the InstructGPT group is mainly due to model introduced text.

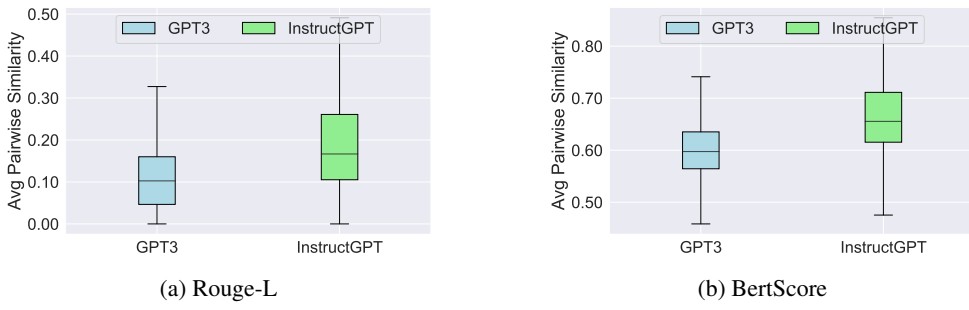

(a) Rouge-L      (b) BertScore

Figure 12: Boxplot of average pairwise similarity of suggestions provided by InstructGPT and GPT3 when calculated using (a) RougeL and (b) BertScore as a measure of similarity. InstructGPT presents more similar suggestions on average to the user by a statistically significant margin.

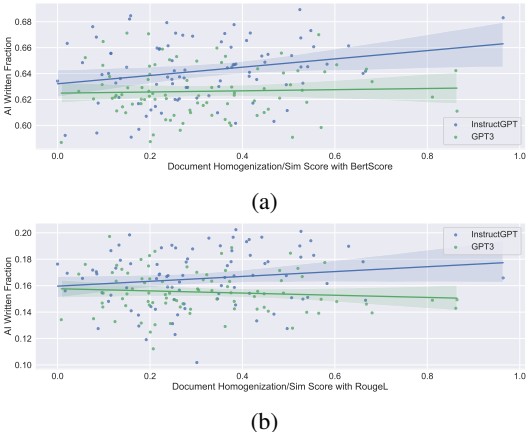

(a)

(b)

Figure 13

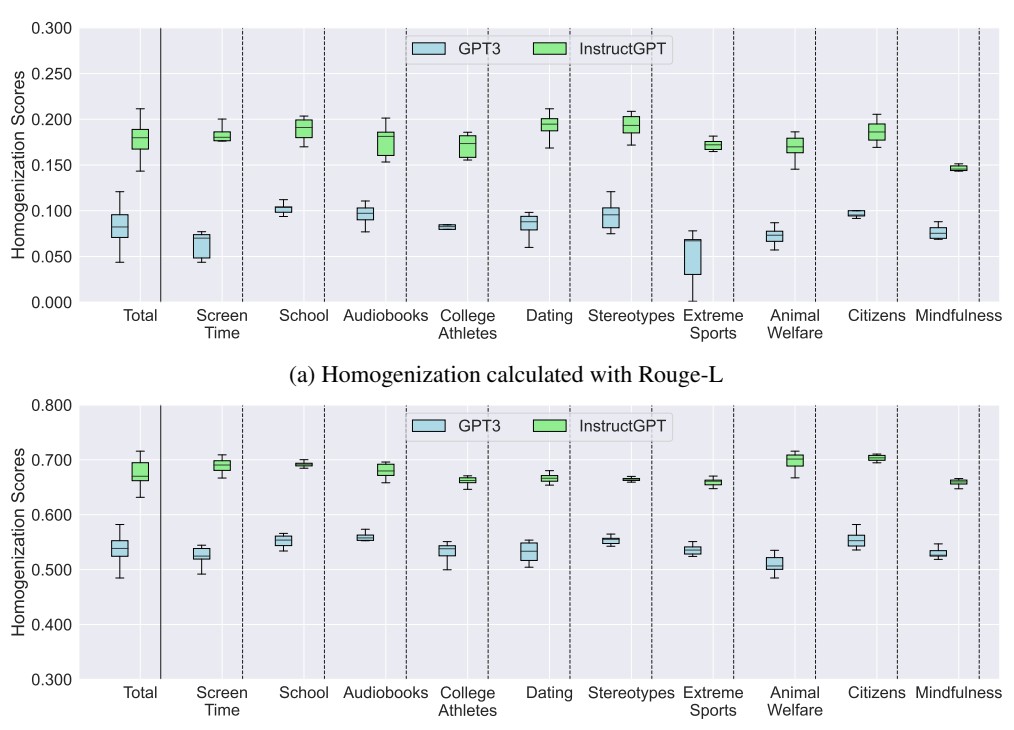

(a) Homogenization calculated with Rouge-L

(b) Homogenization calculated with BertScore

Figure 14: Boxplots of homogenization scores comparing essays generated solely via InstructGPT and GPT3 models at the raw essay level and calculated using (a) Rouge-L and (b) BertScore as measures of similarity. The left-most column (Total) shows essay homogenization scores for all topics and the other columns show essay homogenization scores by topic. Essays written with InstructGPT exhibit higher corpus homogenization by a statistically significant margin.

| $n$-gram size | Solo | GPT3 | InstructGPT |
|---|---|---|---|
| 1 | 0.119 | 0.116 | **0.115** |
| 2 | 0.602 | 0.585 | **0.579** |
| 3 | 0.898 | 0.886 | **0.869** |
| 4 | 0.973 | 0.967 | **0.953** |
| 5 | 0.991 | 0.988 | **0.977** |

Table 11: $N$-gram diversity in each setting. Bold values indicate the lowest diversity score as measured by the fraction of unique $n$-grams. Essays written with InstructGPT are the least diverse across $n$-gram sizes.

| Thresholds | Solo | GPT3 | InstructGPT |
|---|---|---|---|
| 0.5 | 0.982 | 0.971 | **0.950** |
| 0.6 | 0.941 | 0.927 | **0.877** |
| 0.7 | 0.792 | 0.779 | **0.738** |
| 0.8 | 0.543 | 0.514 | **0.494** |

(a) RougeL

| Thresholds | Solo | GPT3 | InstructGPT |
|---|---|---|---|
| 0.1 | 0.998 | 0.997 | **0.992** |
| 0.2 | 0.981 | 0.976 | **0.941** |
| 0.3 | 0.805 | 0.787 | **0.730** |
| 0.4 | 0.321 | 0.338 | **0.292** |

(b) BertScore

Table 12: Diversity measured by agglomerative clustering of the key points of essays with (a) RougeL and (b) BertScore as the metric. We report the scores using different distance thresholds for clustering. Bold values are different to other columns by a statistically significant margin ($p < 0.05$) using a permutation test. InstructGPT consistently exhibits lower content diversity across both similarity metrics and all selected thresholds.

## D LIMITATIONS

**Interaction interface.** Our interface provides suggestions to users in the form of continuations of the current text in the draft. Further investigation is needed to evaluate if the reduction in content diversity with feedback-tuned models can be mitigated with prompt engineering or richer forms of interaction (e.g., through a dialogue).

**User selection.** The outcome of the experiments can be affected by the specific group of participants. We try to ensure a diverse user group and detail our recruitment procedures in Appendix A.1. However, it is unclear whether these results will generalize to other groups such as students learning to write or second language speakers, who have different goals and incentives for using writing assistants.

**LLM access.** Our experiments are conducted using two limited-access models from OpenAI. While we believe they are representative of properties of current LLMs, it is possible that the other models may exhibit different behavior, especially given that the RLHF pipeline is highly customized.

| Algorithm | Solo | GPT3 | InstructGPT |
|---|---|---|---|
| LZMA | 0.305 | 0.302 | **0.293** |
| ZLIB | 0.353 | 0.351 | **0.342** |
| GZIP | 0.352 | 0.341 | **0.350** |

Table 13: Compression ratios of the essays using various lossless compression algorithms. Bold values are different from each of the other columns with statistical significance ($p < 0.05$) using permutation tests. InstructGPT exhibits lower diversity, i.e. lower compression ratios, than GPT3 and InstructGPT across all three compression algorithms.

