# OpenReview forum: "Does Writing with Language Models Reduce Content Diversity?"
_ICLR.cc/2024/Conference — ICLR 2024 poster_

### Official Review · Reviewer_vExG · 2023-10-25

**Soundness:** 3 good
**Presentation:** 4 excellent
**Contribution:** 3 good
**Rating:** 6
**Confidence:** 4

**Summary:**

The paper describes extensive work on the evaluation of language models for content diversity and homogenization. The work defines homogenization as the degree of similarity between the produced content of users co-writing with LLMs measured using overlap metrics such as Rouge-L and BERTScore; while diversity as the degree of linguistic variation of the text measured by n-grams and/or type token ratio (TTR). The authors cite the main motivation of the study being to assess the quantifiable degree of which LLMs reduce content diversity and homogenization in a co-writing scenario. The author/s then conduct a controlled experiment to gather data in 3 setups across several thematic prompts: writing essays w/ LLM assistance (solo), using GPT as is (GPT-3), and using a GPT model finetuned for instruction-following (InstructGPT). From the collected data, the author/s perform several experiments geared towards answering the research questions on (a) how users often engage with the model, (b) homogenization between essays from the three setups, and (c) differences in diversity between the two generative models used in the co-writing scenario. In terms of homogenization, the results of the paper show that using instruction-tuned GPT models like InstructGPT produces much more similar content compared to using these models (obviously) and using regular GPT-3 models without the instruction-tuning. In terms of content diversity, using InstructGPT does have some tendency to reuse word combinations, as evidenced through the n-gram analysis by the authors, and shows a considerable reduction in key-point and lexical (word-based) diversity. Overall, the paper paints a picture of an empirical evidence of how using these models in a co-writing setup affects the quality of texts we co-produce with them.

**Strengths:**

The paper is very well written and structured, and the narrative flows smoothly. I had no problem understanding the picture the author/s are constructing, starting with the problem definition, motivation, methodology, and the discussion of findings. I believe anyone, even a non-technical audience from interdisciplinary fields, will fully understand and appreciate the results of the paper in one sitting.

As highlighted, one of the most important contributions of the paper is the dataset built from the controlled experiment from the three different groups (solo, using GPT3, using InstructGPT). I find value in the availability of this dataset for public use.

I find the quantitative and controlled experiment with how LLMs affect the diversity of texts, especially in a co-writing setting explored in the study, to be extremely useful and timely. It is a common intuition and suspicion in the education community that LLMs do reduce creativity when used for content-generation tasks such as writing essays, but this study fully supports and grounds that intuition with succinct and convincing experiments.

**Weaknesses:**

One thing that feels quite lacking and limited in the paper is that the metric for diversity only focuses on word-based evaluation through n-grams and keypoint analysis. As a reader, I was expecting a more expansive set of linguistic features to be explored to really shed light on the diversity of generated texts from (Instruct)GPT models. This includes syntax or parse tree-based differences and some perspectives even on information theory by measuring the suprisingness (entropy) of human vs machine-generated texts. Well-known previous works in text generation such as (https://arxiv.org/abs/1904.09751) have shown that human-written texts are naturally surprising–which can be a prerequisite for its texts being diverse–therefore exploring this form of feature with the co-writing setup done by the author/s is necessary.

There is a dedicated section for the author/s’ inferences as to why the use of InstructGPT produces less diverse content, but it doesn’t supply the expectations of the reader. I’m quite confused with this takeaway “The reduction in diversity in collaborative writing is attributed more to text contributed directly by the InstructGPT model.” This is an obvious case. Upon reading Section 6, I was expecting more of a deeper discussion as to why InstructGPT reduced diversity more than models not trained for following instructions, such as a regular GPT-3, which can be traced back to some analysis about the quality and characteristics of data used for training these models. In order to level the playing field and to compare these two models equally, you may need to extensively compare their generated content to even more fine-grained linguistic analysis beyond word-based patterns such as n-grams or type-token ratio.

The paper makes heavy use of the Co-Author system by Lee et al (2022), and it seems in the paper that the authors of the system have already done a similar experiment on co-writing with LLMs, especially on the lens of creativity, vocabulary, and grammaticality. The author/s of the submitted paper have only extended this through the addition of homogenization and diversity analysis. I’d like to see more clear and explicit differentiation of the work compared to what analysis has been in the Co-Author paper so that it does not seem to be incremental. I invite the authors to discuss this.

**Questions:**

1. Once the essay has been completed, were the users required to reread and revise the essay for another round? This may negatively bias the result with the n-gram pattern analysis as sometimes users may freely switch words for more fitting synonyms in phrases while the GPT-generated remains constant.

2. Are the writers recruited classified as native speakers of English? There are multiple claims in literature (https://arxiv.org/abs/2304.02819, https://hai.stanford.edu/news/ai-detectors-biased-against-non-native-english-writers, https://arxiv.org/abs/2306.04723) stating that there are similarities with writing styles of non-native speakers of English with machine-generated texts. Recruiting non-native English speakers in the mix may introduce some bias.

3. Why is the model-suggested text not required for groups using GPT models? Does this mean that there are possibilities that some passages in those two groups (users using GPT and InstructGPT) have fully human-written text since they are not required to use the model generations anyway?

4. For writers that are in the group using LLMs, is there a checker to check whether their outputs are a certain percentage of human and LLM-generated content? If a writer fully made use of the LLM generations and then only contributed one to two sentences, would this be considered a qualified entry for analysis? I understand that the experiment is on co-writing but it would be pointless then if more than 80% of the submitted text passage has been written by GPT.

5. Is there a possible method that you can integrate to quantify the usable words forming the overall vocabulary for each language model? I believe this would greatly support the claim that using LLMs may reduce diversity and creative in writing. In addition, what does the literature say about the quantifiable reduction of the cost of instruction-tuning LLMs to the degradation of its diversity? The papers made a mention of this but only at a surface-level detail without going deeper into its discussion.

---

> ### Author Response · Authors · 2023-11-18
>
> (1/2)
> Thank you for taking the time to review our paper and providing valuable feedback and questions. We appreciate your comments on the timeliness and presentation of our paper and the value of collecting and releasing the dataset of writing logs + essays to facilitate research in the area.
>
> **To address the concern about the comprehensiveness of reporting metrics studying the linguistic aspects of the collected essays:**
> We used metrics based on n-grams and key points to measure content diversity in an interpretable way. We acknowledge that including syntactic and information-theoretic metrics would lead to a more holistic analysis of the collected essays and hence we will include the following summary statistics. We add these to Table 9 in Appendix C. We would like to highlight that releasing the essays collected would allow further in-depth analysis of the same.
> 1) Predictably, the essays written with model help have a lower average perplexity (or entropy) as measured via GPT2. This effect is more pronounced with InstructGPT than GPT3, complementing the results on n-gram and key point diversity, again highlighting that writing with model help can result in more generic content.
> 2) We find that the essays from each group are roughly the same length. When writing with InstructGPT, we find that users write slightly shorter sentences. Additionally, by calculating the average height of the dependency parse trees of the sentences in the essays, we observe that writing with InstructGPT results in sentences with fewer nested structures, indicative of slightly lower complexity.
> 3) Writing with model help also results in fewer unique POS-Ngrams again providing some evidence that collaborative writing could result in more homogenized usage of language. The effect here is even between GPT3 and InstructGPT, a slight departure from the results on diversity which is more focused on content. We finally plot the distributions of the 50 most frequent POS-Ngrams and find that on 4 and 5-grams, writing with model help leads to higher repetition of frequent POS-Ngrams.
>
> Results in markdown (Same as Table 9 in the paper, numbers in brackets indicate standard errors on the computed average values):
> |                                 |                     |      Solo     |      GPT3     |  InstructGPT  |
> |---------------------------------|---------------------|:-------------:|:-------------:|:-------------:|
> | Perplexity of Essays (via GPT2) |                     |        25.067 |         22.10 |         20.26 |
> |      Sentence Length (in words) |                     |  23.51 (0.25) |  23.66 (0.24) |  22.51 (0.24) |
> |           Height of Syntax Tree |                     |   5.93 (0.06) |   5.98 (0.06) |   5.71 (0.05) |
> |         Essay Length (in words) |                     | 376.44 (8.03) | 380.87 (9.30) | 368.39 (8.43) |
> |                   Human Ratings | Relevance to Prompt |          4.30 |          4.15 |          4.10 |
> |                                 |      Grammaticality |          4.00 |          4.10 |          4.00 |
> |                                 | Depth of Discussion |          3.95 |          3.85 |          3.80 |
> |               Unique POS Ngrams |                   1 |            73 |            56 |            58 |
> |                                 |                   2 |           487 |           437 |           426 |
> |                                 |                   3 |          1297 |          1261 |          1235 |
> |                                 |                   4 |          2044 |          1975 |          1988 |
> |                                 |                   5 |          2423 |          2338 |          2414 |
>
> **Regarding the differences to CoAuthor:**
> 1) While CoAuthor presented an observational study on the users’ writing processes with model help, we aim to establish the relationship between the intervention of model help and the diversity of content produced. As a result, we collect not only collaborative essays using different models but **also essays written by the same authors without model help (which is missing in CoAuthor)**. CoAuthor primarily analyzed the impact of the sampling temperature on various metrics such as vocabulary and machine-contribution.
> 2) We also introduce the key point-based metric to better capture the content diversity between the setups. We believe that as research on the societal implications of introducing models in public-facing settings progresses, we would need to go beyond just traditional metrics in the literature measuring grammaticality and perplexity and our work provides a new axis of LLM evaluation in these real-world settings.

---

> ### Author Response · Authors · 2023-11-18
>
> (2/2)
>
> **Addressing the concern that the reduction in diversity is attributed to the text contributed by InstructGPT is obvious:**
> While the high-level finding might align with the view of researchers, in the collaborative setting it is not necessarily clear that the model-generated text is the only possible reason for these effects as recent work has also shown that interacting with the models might result in the users altering their behavior [1]. As a result, we compare the key points introduced by the humans and models in both the GPT3 and InstructGPT setups and find that the human contribution in both setups is almost identical, while the model-attributed key points cause the difference in diversity.
>
> [1] Maurice Jakesch, Advait Bhat, Daniel Buschek, Lior Zalmanson, and Mor Naaman. 2023. Co-Writing with Opinionated Language Models Affects Users’ Views. In Proceedings of the 2023 CHI Conference on Human Factors in Computing Systems (CHI '23). Association for Computing Machinery, New York, NY, USA, Article 111, 1–15. https://doi.org/10.1145/3544548.3581196
>
> **To answer the questions:**
> **Q1) Were the users instructed to read and revise their essays?** Users are allowed to edit their essays post-completion. Their objective is to convey their opinions effectively. We acknowledge that this might affect word usage patterns which again motivates the contribution of our proposed key point-based evaluation as these key points are less likely to be changed during post-editing.
>
> **Q2) Are the recruited writers native English speakers?** The users in this study are indeed largely native English speakers from the United States recruited from Upwork.
>
>
> **Q3 and 4) Why was introducing model-generated text not enforced for the group writing with model help? And relatedly, what percentage of essays were written by humans?** We don’t enforce that users include a certain amount of machine-generated text as we want to study how collaborative writing in its most natural form differs from solo writing without model help. The average machine-written fraction is 0.32 for InstructGPT and 0.35 for GPT3 (Table 1) indicating that users obtain useful help from the models without over-relying on the same. The vast majority of essays have machine-written fractions ranging from 0.2 to 0.5 with a few outliers being as low as 0.1 and as high as 0.8 (Figure 12 in Appendix C).
>
> **Q5) Can we quantify the fraction of the tokenizer used in the generated text? What does prior work discuss about the reduction in diversity due to instruction tuning?** The usage of the overall vocabulary i.e. the fraction of words from the tokenizer that are included in the generated text could be an interesting signal of creativity, however, the measure we would be really interested in is actually the fraction of words that are also relevant to the topic being written about. This is potentially more challenging to calculate. We do note that the data we collect, including all generated model suggestions is released publicly enabling further analysis along these lines. The specific findings discussed in prior work about the reduction in diversity include the reduction in the entropy of the output distribution after training with RLHF (Figure 15 in [2]) and the observation that instruction tuning reduces the calibration of the GPT-4  (Figure 8 in [1]).
>
> [1] OpenAI. 2023. Gpt-4 technical report.
>
> [2] Yuntao Bai, Andy Jones, Kamal Ndousse, Amanda Askell, Anna Chen, Nova DasSarma, Dawn Drain, Stanislav Fort, Deep Ganguli, Tom Henighan, et al. 2022. Training a helpful and harmless assistant with reinforcement learning from human feedback. arXiv preprint arXiv:2204.05862

---

> > ### Comment · Reviewer_vExG · 2023-11-21
> > **Acknowledge additional results and author response**
> >
> > This is to confirm that I have read and considered my fellow reviewer's feedback and the author/s response. My questions have been answered succinctly but the given response should be added to the paper itself particularly on important details such as the (a) setup of the experiments, (b) clear distinction in the setup and collected data between the paper and the CoAuthor work, as well as the details about the workers. I also find great value from Reviewer 1's feedback which should be incorporated into future iterations of the paper.
> >
> > I'm increasing my Overall score by 1 as well as the Contribution score.

---

> > > ### Author Response · Authors · 2023-11-22
> > >
> > > We appreciate your revision. To summarize the updates to the paper thus far, we have included the details regarding the significance testing, recruited users, and instructions to participants (Section 4.2, Appendix A.1, B.3) as well as the differentiation to CoAuthor (Section 7). We intend to continue updating the draft as we receive feedback while remaining within the page limit.

---

### Official Review · Reviewer_qs9Z · 2023-10-25

**Soundness:** 3 good
**Presentation:** 3 good
**Contribution:** 3 good
**Rating:** 6
**Confidence:** 4

**Summary:**

The authors measure the diversity of outputs in LLM-assisted essay writing. They compare three essay setups: written solely by a user, written by a user + gpt3, and written by a user + instruct-gpt. They find that essays written using instruct-gpt are significantly less diverse than the others. Disaggregating the text by LLM-authored or user-authored shows that this lack of diversity comes from the model itself (ie, the user is *not* more repetitive when using instruct-gpt). They also present a novel diversity metric (by first summarizing the essay into "key points") to measure the diversity of the essay on a structural level.

**Strengths:**

- It sometimes feels like there's a mismatch between how the academic world studies and evaluates LLMs, and how LLMs are actually used in the real world. This paper bridges that gap, and directly asks important questions that are relevant to today's users of ChatGPT.
- It's interesting that you use the LLMs as part of the process in generating the key points .
- The diversity metric of "key points" that the authors present is useful and interesting-- as LLMs become ubiquitous, it's clear that measures of text diversity/similarity/etc are often insufficient, so this type of work can have a significant impact.

**Weaknesses:**

- If we know that instruct-gpt generates lower-diversity text, then I think I am missing a bit of why this is a surprising result overall.
- I'm not convinced that these results can be extrapolated to other LLM-based writing tools. When designing a good tool, questions like "is it limiting what the writer wants to express" are some of the things you'd first think about addressing. In fact, other work has found that having the LLM can be helpful for getting users out of their normal writing patterns [(e.g.)](https://wordcraft-writers-workshop.appspot.com/learn). But then again, ChatGPT is so ubiquitous that maybe it's enough to just evaluate it alone.
- Having a precise character number on the text written by the model vs the user can be misleading. For example, what if the model generates a sentence, and the user fully rewrites it but keeps the underlying meaning? (having worked on similar tools, this seems to happen frequently)
- Nit: Figure 4a-- it's confusing that the "InstructGPT" and "GPT3" color key is horizontal and lines up with the vertical center division line (I thought maybe GPT3 was the label for the left half or something). Maybe just put them on top of each other like 4b?
- It would be good to compare to essays written entirely with instruct-gpt and GPT3
- Figure 4b is generally kind of confusing, especially the x axis. Can you just simplify and say "average probability for top 10 ngrams" or something? (This is just a suggestion-- I'm not sure it's actually better. But the current version could use more explanation).

**Questions:**

- If the key points are generated with GPT3.5, aren't they inherently more likely to be similar to other text generated with GPT3 or ChatGPT than with human text? If so, won't that skew the results?
- (More questions in "weaknesses")

---

> ### Author Response · Authors · 2023-11-18
>
> (1/2)
>
> Thank you for taking the time to review our paper and providing valuable feedback and questions. We appreciate your comments on the value of connecting LLM evals in research to real-world scenarios as well as using LLMs to create higher-quality metrics for measuring complex effects such as content diversity.
>
> **Regarding the surprisingness of the results:** While InstructGPT has been shown to produce less diverse text than GPT3 (corroborated by our observations in Figure 12 and Section 6), in the collaborative setting human intervention (e.g., they can edit or reject model suggestions) may mitigate this impact of the model. Indeed despite only 32% of the essays contributed by InstructGPT and 35% by GPT3, or around 66% of the essays written by the human users, we observe that the contribution of InstructGPT is sufficient to reduce the overall diversity of the generated essays.  While this might agree with the intuitions of researchers, we believe the value of our findings lies in quantifying and measuring this effect rigorously, given the unclear effect of human interaction in the collaborative process. By making the collected data public, we hope to foster further research along these directions.
>
> **To address the concern about the generalizability of our findings to other LLM-based writing tools.**
> While our results focus on a set of OpenAI models, we believe they hold significance due to two key reasons. Firstly, many tools released by various companies are built upon OpenAI APIs as they are recognized for their high performance (https://crfm.stanford.edu/helm/latest/?group=core_scenarios) and usability in products. Secondly, our chosen models enable a comparison between a general-purpose base LLM (davinci) and an RLHF-trained equivalent (text-davinci-003), allowing us to generalize the distinction in training methods to other model families.
> Additionally, in our collaborative writing setup, the continuations are sampled from the model given the current user draft as the prompt i.e. we can test the model impact averaging over many different prompts, again contributing to the generalizability of these results. Finally, to address the aforementioned findings from the WordCraft project that LLMs can help dislodge writers from their common patterns, we believe that our results complement their findings and highlight a potential concern as well. Consider the scenario where a professional author is stuck while writing and accepts model suggestions to dislodge themselves from their block. While that writer’s productivity improves, it becomes a concern if similar ideas to shift out of a block are used by many different users. Our findings suggest that this improvement in individual productivity could potentially arise at the cost of a homogenizing effect, resulting in less colorful stories in the long run. In the WordCraft paper itself, it was noted that the model often resorted to standard tropes as suggestions. This motivates why we chose to study the effect of model intervention in aggregate across different users and introduce quantitative metrics to observe the reduction in diversity.
>
> **Clarifying how the fraction of model intervention is calculated:** We clarify that if a model generates a sentence and the user fully rewrites that sentence, then the machine-written fraction here would be zero. Our text editor interface tracks if each character was most recently introduced by the model generation or by the user’s keystrokes by recording each change of character in the draft. This is what we use to compute the model contribution and attribute key points to the model and user which we use for analysis in Section 6. While this does not include the contribution of the user merely seeing the model suggestions and then paraphrasing these into the draft, we do record every single suggestion shown to the user and this can be matched up to the final artifact produced by the user. Hence we again believe that our interface and collected data could serve as a valuable resource to the community.

---

> ### Author Response · Authors · 2023-11-18
>
> (2/2)
>
> **We compare the homogenization of essays generated by GPT3 and InstructGPT without any human intervention** in the Figure 14 in the updated draft. We prompt the models to collect the essays with 10 variants of the following sentence ‘Write a three to four-paragraph argumentative essay on this prompt in the first person: + {Essay Prompt}’. We collect 10 essays on 5 topics, sampled with a high temperature of 0.9, and report the homogenization scores below. Overall the trend shows that essays generated with InstructGPT show higher homogenization across all topics, with both models showcasing less variance within each setup than the collaborative experiments with human users. While not directly comparable to the Solo or collaborative essays due to the different way in which the data was collected, these results further motivate our analysis from Section 6 that the contribution from the models is impacting the diversity and not a change in human behavior in the presence of the model.
>
> **We’ve updated Figure 4a per your feedback.**
>
> **Q1: If we use GPT-3.5 to summarize the essays for key points, will the model-generated text be more similar than the human-written text.**
> While the key points are generated by GPT 3.5, this effect is even across all setups i.e. we use the same GPT-3.5 model to summarize the essays even from the Solo-writing format. As the same model is used to summarize key points from each setup, the effect is averaged out and does not impact the results themselves.

---

> ### Author Response · Authors · 2023-11-22
>
> Thanks a lot for your extensive review. As the discussion period is nearing its end, please let us know if you have any remaining questions. We hope that we were able to convey the novelty of our findings and clarify your other questions. We will continue to update future versions of the paper based on your feedback!

---

### Official Review · Reviewer_ttyL · 2023-10-31

**Soundness:** 2 fair
**Presentation:** 3 good
**Contribution:** 2 fair
**Rating:** 5
**Confidence:** 3

**Summary:**

This paper investigates whether writing assisted by large language models (LLMs) leads to reduced diversity in the composed text. It uses a within-subject design (but see my comment about the confusion below) that includes three groups of participants that wrote essays without using an LLM, a group that used a base LLM (GPT3) to assist writing, and a group that used an LLM tuned with human feedback (InstructGPT) to assist writing. The main finding of the paper is that writing with InstructGPT leads to significantly more homogeneous context and less diverse n-gram (n>1) distribution and semantic key points (as determined by prompting gpt-3.5-turbo) than writing without an LLM and writing with the GPT3 base LLM. The authors further attributed the increased homogeneity and reduced diversity to the portions of text contributed by the LLM (instead of those portions contributed by the human).

**Strengths:**

S1. Taking an empirical approach to answer the interesting and important question about how humans interact with LLMs when composing text. This is a first investigation in this topic area to my knowledge.
S2. Defining a number of metrics to quantify the diversity and homogeneity of text written under different human-LLM collaboration settings, encompassing both the lexical aspect (n-grams) and semantic ones (key points extracted by gpt-3.5 and BertScore based similarity).

**Weaknesses:**

W1. There are some confusion / lack of clarity about the experiment design. While Appendix Sections A.1 and A.3 clearly indicate that the authors used a within-subject design wherein each participant wrote text under all three settings, the text on p. 1 ("control group", "LLM treatment group", and "feedback-tuned LLM treatment group") seems to indicate that the study was based on a between-subject design, where nonoverlapping sets of participants were tested under the three different settings. This confusion is further acerbated by the fact that the authors used independent-samples t-test for statistical analysis throughout this manuscript, which indicates this is a between-subject design. If the study was really based on a within-subject design, as Sections A.1 and A.3 indicate, then the statistics should be redone with repeated-measures ANOVA and paired t-tests, which may alter the significance test results of this paper.
W2. While reading this manuscript, I can't help but feel some duplication between Sections 4 and 5, which focuses on "homogeneity" and "diversity", respectively. Although these two terms usually refer to slightly different aspects of a collection of text, the authors never tried to define them and how they are related to and different from each other in this paper. So in the end, the reader has to read and understand that "homogeneity" is operationally defined as the average of a semantic distance metric (e.g., BertScore) among pairs of the composed text in the corpus, while diversity refers to both lexical and semantic aspects, and are calculated in a different way (# of unique lexical or semantic units).
W3. The user interaction paradigm is limited to generating and adopting continuations with the LLM (whether base or feedback-tuned), as described throughout the paper and in Section A.3. However, anecdotally we know that users of LLM and associated chatbots (e.g., ChatGPT and Bard) often prompt the models to write or rewrite text by using their own prompt templates or instructions. The flexibility is higher in such prompting-based interactions than in the more stereotypical continuation paradigm. The authors did not mention such instruction- and prompting-based interaction paradigms, which are arguably more prevalent and useful compared to simple continuation.
W4. To evaluate the text composed by the study participants, the authors used only homogeneity and diversity metrics. They didn't report more basic metrics of the composed texts, which IMO should also be reported. These include the length, the perplexity (e.g., as determined by GPT3), and writing quality (e.g., grammaticality and other human rater-based writing quality) metrics. These basic metrics are not only important for ensuring comprehensiveness of the result reporting, but also useful for ruling out the possibility that differences in homogeneity and diversity are somehow due to the confounding factors due to differences in those basic aspects of writing.

**Questions:**

Q1. See my comments in W1 above. Please clarify whether the study design was between-subject or within-subject. Please ensure that the proper statistical analysis is used accordingly.
Q2. Were the base LLM (davinci) and feedback-tuned LLM (text-da-vinci-003) based on the same model architecture and the same parameter count to make it a fair comparison between the texts composed with them?

---

> ### Author Response · Authors · 2023-11-18
>
> (1/2)
>
> Thank you for taking the time to review our paper and providing valuable feedback and questions. We appreciate your comments on the importance of defining and quantifying metrics for measuring the effect of model intervention on content diversity. We discuss some of your concerns below
>
> **W1: To clarify the experimental setup and address the concern that an independent samples t-test might not be appropriate for the chosen experiment design:**
> We recruited a fixed set of users who wrote essays in each of the three setups in a randomized order. We ensure that they write essays on different topics in each setup to prevent a repetition of their opinions. However, this doesn’t fall clearly into the within-subjects study as the topic which they write on is different, thus we cannot create pairs of essays written by the same authors in different setups since these will be on different topics (which is a confounder). As a result, we performed significance testing via independent t-tests.
>
> To address your concerns regarding statistical significance, we observe that within each topic, each writer only submits an essay to one of the three setups. This matches the between-group setup, albeit with less power as we only collected 10 essays per setup per topic. We compute the significance of the difference in homogenization scores per topic via an independent samples t-test. The p-values are reported in tables 7 and 8 of the draft comparing each pair of setups within each topic at both the key point and essay level. Comparisons with significance at the 5% level are marked with an asterisk. We observe that the InstructGPT setup exhibits higher homogenization over Solo writers with significance at the 5% level on 8 out of 10 topics at the raw essay level and 6 out of 10 at the key point level using both Rouge-L and BertScore. The same trend holds on GPT3 on 7 topics at the raw essay level and 5 at the key point level.
> We believe that this confirms the overall trend that writing with InstructGPT results in higher homogenization in the essays created and appreciate your note to improve the rigor of our testing.
> We have added this to the paper in Appendix B.3, tables 7 and 8 including a forward reference to these results in Section 4.2.
>
> **W2: Addressing the distinction between concepts of homogeneity and diversity being measured in the paper:**
> In general, homogenization measures whether the outputs from a person are similar to the outputs of another (or group of people) while diversity measures how much redundancy there are in a set of outputs from a group of people. As noted, these concerns are related whereby high homogeneity (i.e. documents are quite similar to one another) between essays would correlate with lower diversity (i.e. key points made across documents are repeated). From a practitioner’s perspective, if you are interested in the authenticity of the text, then you would want to measure homogeneity which indicates whether the author’s voice is lost during co-writing. If you want to study the aggregate property of text, diversity measures how language as a whole changes when the writing process is intervened by LLMs.
>
> **W3: Discussing how variations in prompting strategies would impact the results of our study:**
> While we do not investigate the role of the formatting of prompts, in this work since we sample continuations from the models for the current draft, the essays as written by different users serve as the prompts to the model for each query. However, the homogenization and diversity effects we observe occur despite this variance in prompting. A deeper exploration into how humans would prompt models to potentially mitigate this issue is a direction of future research, although it is unclear how different the prompts generated by different users would be as recent findings have shown that users do not systematically explore methods of prompting [1].
>
> [1] J.D. Zamfirescu-Pereira, Richmond Y. Wong, Bjoern Hartmann, and Qian Yang. 2023. Why Johnny Can’t Prompt: How Non-AI Experts Try (and Fail) to Design LLM Prompts. In Proceedings of the 2023 CHI Conference on Human Factors in Computing Systems (CHI '23). Association for Computing Machinery, New York, NY, USA, Article 437, 1–21. https://doi.org/10.1145/3544548.3581388

---

> ### Author Response · Authors · 2023-11-18
>
> (2/2)
>
> **W4: To address the concern about simple confounding factors to the observed diversity effects that could be addressed by reporting more statistics about the collected essays:**
> We aim to improve the comprehensiveness of our analysis by reporting additional metrics on the collected essays as follows. We add these to Table 9 in Appendix C.:
> 1) We computed basic statistics such as perplexity, average sentence length, and essay length on the essays collected. Predictably, writing with model-help reduces the perplexity of the essays which now incorporate suggestions sampled from an LM distribution. The total length of essays, in terms of word count, is roughly similar, with writers writing slightly shorter sentences when writing with InstructGPT (albeit with a slightly high standard error).
> 2) To ensure that the quality of the essays is not a confounder in the result, we present human ratings for a random sample of 10 essays (one per topic) from each setup that we conducted as part of our quality control checks. Here we judged the essays on 3 different axes - relevance to the topic, grammaticality, and the depth of discussion on their opinions. This was per the advice from faculty at the Expository Writing program at our university. This analysis showed no clear differences between the setups, explained in part due to the observation that the machine-written fraction in the collaborative essays was on average around 32%-35% (Table 1).
>
> Results in markdown (numbers in brackets refer to the standard errors on calculating the average values in the cells):
>
> |                                 |                     |      Solo     |      GPT3     |  InstructGPT  |
> |---------------------------------|---------------------|:-------------:|:-------------:|:-------------:|
> | Perplexity of Essays (via GPT2) |                     |        25.067 |         22.10 |         20.26 |
> |      Sentence Length (in words) |                     |  23.51 (0.25) |  23.66 (0.24) |  22.51 (0.24) |
> |           Height of Syntax Tree |                     |   5.93 (0.06) |   5.98 (0.06) |   5.71 (0.05) |
> |         Essay Length (in words) |                     | 376.44 (8.03) | 380.87 (9.30) | 368.39 (8.43) |
> |                   Human Ratings | Relevance to Prompt |          4.30 |          4.15 |          4.10 |
> |                                 |      Grammaticality |          4.00 |          4.10 |          4.00 |
> |                                 | Depth of Discussion |          3.95 |          3.85 |          3.80 |
> |               Unique POS Ngrams |                   1 |            73 |            56 |            58 |
> |                                 |                   2 |           487 |           437 |           426 |
> |                                 |                   3 |          1297 |          1261 |          1235 |
> |                                 |                   4 |          2044 |          1975 |          1988 |
> |                                 |                   5 |          2423 |          2338 |          2414 |
>
>
> **Q2: Clarification that the model sizes of the chosen base LLM (davinci) and RLHF model (text-davinci-003) are comparable:**
> The OpenAI model index (https://platform.openai.com/docs/model-index-for-researchers) confirms that the davinci endpoint was the 175B parameter model released with the original GPT3 paper i.e. a pure LLM, and that text-davinci-003 was of the same size with subsequent training with RLHF (PPO).

---

> ### Author Response · Authors · 2023-11-22
>
> Thanks a lot for your extensive review. As the discussion period is nearing its end, please let us know if you have any remaining questions. We hope that the additional results analyzing the essays and statistical tests presented convey the significance of the results and will continue to update future versions of the paper based on your feedback!

---

> ### Comment · Reviewer_ttyL · 2023-11-23
> **Response to authors comments**
>
> I thank the authors for their reply to my W1. Now I understand why the authors use independent-sample t-tests despite that the same group of subjects were used in the three different tasks. I agree with the authors that due to the differences in the topics, this is better treated as a between-subject design.
>
> Regarding W2, while the authors' reply helped a little, I still feel the two concepts ("homogeneity" and "diversity") will be confusing to readers. I trust the authors will make an effort to clarify this as much as possible in revisions.
>
> As for W3, thanks for acknowledging that users' own instruction and prompting during LLM-assisted text composing is a direction for future research (and should be mentioned in the revised paper). I agree that not directly addressing instruction and prompting in the current paper is fine.
>
> As for W4, the table looks informative and comprehensive. I appreciate the authors effort in performing the additional analysis.
>
> As such, I'm willing to updating my rating from  "5: marginally below the acceptance threshold" to "6: marginally above the acceptance threshold" if that's allowed by the ICLR review process.

---

> > ### Author Response · Authors · 2023-11-23
> >
> > Thank you for your update! We will continue to work with any feedback we receive to keep updating the paper.

---

### Author Response · Authors · 2023-11-18
**Summarizing our rebuttal addressing reviewer concerns**

We thank the reviewers for their feedback. We are grateful that they appreciated our contributions in empirically defining metrics for measuring content diversity (ttyL), utilizing LLMs in the pipeline to formulate usable measures in the real-world setting (qs9Z), releasing the dataset for reuse by the research community (vExG).

To summarize our response, we provide additional analysis of the collected essays using syntax-based measures (vExG) and human judgments (ttyL), and further statistical tests to showcase the significance of the findings (ttyL). We also highlight the novelty of our findings (qs9Z) and differentiate our research from prior work (vExG).

In the revised paper, we include these additional findings from statistical tests and analysis of the essays in Tables 7,8 and 9 as well as Figure 6. We describe these in Appendix B.3 and C adding forward references in the main body of the paper. For now, we highlight these changes in green which will be reverted before the Camera-Ready submission if selected.

---

### Meta-Review · Area_Chair_eoyF · 2023-12-10

**Metareview:**

This paper presents results of a systematic study of the use of LLMs for writing. The research question asks whether the LLM-assistance results in more homogeneous and less diverse essays, and the results show that instruction-tuned LLMs indeed lead to less diversity. One noteworthy contribution is the key point-based metric which seems to be more meaningful than just n-gram overlap for measuring diversity in essays. There were some concerns and questions about the experimental design and comparisons with existing work, but the author rebuttal addressed all of them sufficiently well. Overall, the research question is important and timely, the approach including the metric is novel, and the empirical study was conducted rigorously.

**Justification For Why Not Higher Score:**

The novelty is somewhat limited, and the interest level of the community may be comparatively low.

**Justification For Why Not Lower Score:**

This is an interesting and timely topic, and the study was conducted quite well.

---

### Decision · Program_Chairs · 2024-01-16

Accept (poster)